# IRF2 deficiency disrupts pyroptosis, NK cell interferon-γ production and resistance to *Francisella*

Maxence Cornut[1], Sophia Djebali [1,2,4], Elena Rondeau[1,4], Sarah Dayet[1], Théo Fayolle[1], Julie Haagen[1], Lucie Fallone[1], Noémi Rousseaux[1], Emmanuelle Caspar[1], Mélissa Marcotte[1], Amandine Martin[1], Elise Courteboeuf[1], Maëlan Deschamps-Biboulet[1], Marie Teixeira[1,3], Jacqueline Marvel[1], Bénédicte F Py[1], Thierry Walzer [1], Antoine Marcais[1], Thomas Henry [1,5]✉ & Émilie Bourdonnay [1,5]✉

## Abstract

**IRF2 plays an indirect role in inflammasome activation by regulating Caspase-4 and Gasdermin D (GSDMD) levels. However, the in vivo relevance of this regulatory circuit is unknown. We generate IRF2^KO mice and demonstrate that they are equally susceptible to *Francisella novicida* infection as GSDMD^KO mice. Interestingly, the phenotypes of IRF2^KO and GSDMD^KO mice diverge with respect to IFN-γ. Specifically, IRF2^KO mice exhibit a profound defect in IFN-γ production, which we attribute to an intrinsic role of IRF2 in regulating both the number and maturation of NK cells. IRF2^KO NK cells fail to express the antibacterial effectors IL-18R and Granzyme A, thereby impairing bacterial clearance. IFN-γ therapy partially restores immune responses in IRF2^KO mice and resistance to infection. These findings confirm IRF2 as a dual regulator of inflammasome activity and NK cell function, highlighting its pivotal role in innate immunity. Moreover, they underscore the potential of IFN-γ therapy as a promising treatment for severe infections in patients with primary immunodeficiencies affecting multiple immune pathways.**

**Keywords** IRF2; Gasdermin D; NK Cells; IFN- γ; *Francisella novicida*
**Subject Categories** Immunology; Microbiology, Virology & Host Pathogen Interaction; Signal Transduction

## Introduction

The Gram-negative bacterium *Francisella tularensis* is responsible for tularemia, a zoonotic illness. The severity of the disease varies depending on the route of entry, infectious dose, and subspecies of the infecting strain. *F. tularensis* is a facultative intracellular pathogen replicating in phagocytes. Following phagocytosis, *F. tularensis* escapes from the phagosome into the host cytosol using a type VI secretion system (T6SS) encoded within the *Francisella* Pathogenicity Island (FPI) (Nano et al, 2004; Lindgren et al, 2004; Broms et al, 2010; Valeva et al, 2023). Once inside the cytosol, *F. tularensis* replicates to very high levels. The T6SS is the major virulence factor of *F. tularensis*, and mutant strains lacking the FPI (ΔFPI) are avirulent in mice (Nano et al, 2004; Barker et al, 2009).

*F. tularensis* ssp. *novicida*, also known as *F. novicida* (Johansson et al, 2010), is a bacterium that is avirulent for immunocompetent humans but is highly virulent in mouse models of tularemia. Immune responses to *F. novicida* are dependent on the AIM2 inflammasome (Fernandes-Alnemri et al, 2010; Belhocine and Monack, 2012; Meunier et al, 2015). Activated caspase-1 within the AIM2 inflammasome cleaves pro-IL-1β and pro-IL-18, generating their active forms IL-1β and IL-18. Caspase-1 also cleaves Gasdermin D (GSDMD), producing an N-terminal fragment that forms pores in the host cell membrane. These GSDMD pores enable the release of mature IL-1β and IL-18 and drive pyroptosis, an inflammatory form of cell death that restricts bacterial replication (Zhu et al, 2018; Shi et al, 2015; He et al, 2015). IL-18 promotes IFN-γ production by NK cells during the early phase of *F. novicida* infection (Pierini et al, 2013). Cytokines and GSDMD-dependent responses contribute to host defense to *F. novicida*, although side-by-side comparisons of IL-1R/IL-18R^DKO mice and GSDMD^KO mice suggest that cytokines are more important than GSDMD to promote the survival of mice to *F. novicida* infection (Zhu et al, 2018; Mariathasan et al, 2005).

Alongside this canonical, caspase-1-dependent inflammasome pathway, the non-canonical caspase-11-dependent inflammasome (and its human orthologs, caspase-4 and -5) directly senses cytosolic lipopolysaccharide (LPS) and cleaves GSDMD to trigger pyroptosis (Shi et al, 2015; Kayagaki et al, 2015; Yang et al, 2015; Vak et al, 2019). *F. novicida*, due to its tetra-acylated LPS, escapes caspase-11 detection (Hagar et al, 2013).

The transcription factor IRF2 was identified through a genome-wide CRISPR-Cas9 screen as a regulator of caspase-4 in human cells (Benaoudia et al, 2019). It was also identified through a

[1]CIRI, Centre International de Recherche en Infectiologie, Université Claude Bernard Lyon 1, Inserm U1111, CNRS, UMR5308, ENS de Lyon, F-69007 Lyon, France. [2]SFR Biosciences_AniRA ImmOs (Université Claude Bernard Lyon 1, CNRS UAR3444, Inserm US8, ENS de Lyon, AniRA-ImmOs, LYON, France. [3]SFR Biosciences_AniRA PBES (Université Claude Bernard Lyon 1, CNRS UAR3444, Inserm US8, ENS de Lyon, AniRA-PBES, LYON, France. [4]These authors contributed equally: Sophia Djebali, Elena Rondeau. [5]These authors jointly supervised this work: Thomas Henry, Émilie Bourdonnay. ✉E-mail: thomas.henry@inserm.fr; emilie.bourdonnay@inserm.fr

chemical screen as a regulator of GSDMD expression in murine macrophages and certain human cells (Kayagaki et al, 2019; Thygesen and Stacey, 2019). While these findings suggest a pivotal role for IRF2 as a positive regulator of inflammasomes, notably via GSDMD regulation, the relevance of this regulation remains to be demonstrated in vivo. The present study aimed to address this gap in knowledge by examining the involvement of IRF2 in *F. novicida*-induced inflammasome activation and, more globally, in the antibacterial immune responses in mice.

# Results

## IRF2$^{KO}$ mice exhibit high susceptibility to *Francisella novicida* infection, correlating with a strong and selective impairment in IFN-γ production

To explore the in vivo role of IRF2 in the immune responses against *F. novicida*, we generated IRF2$^{KO}$ mice using CRISPR/Cas9 genome editing (Fig. EV1A,B).

Successful *IRF2* deletion was confirmed by immunoblotting (Fig. EV1C), while IRF1 protein levels remained unchanged in the absence of *IRF2* (Fig. EV1D–F). These findings confirm that IRF2 deficiency does not trigger compensatory IRF1 expression, supporting the conclusion that the observed phenotypes are directly attributable to IRF2 loss.

We infected them and their co-housed WT littermates, subcutaneously with *F. novicida*. All IRF2$^{KO}$ mice succumbed within 4 days post-inoculation, significantly before WT mice, of which 45% survived until the end of the experiment (Fig. 1A). Upon systemic dissemination, higher colony-forming units (CFU) were observed in the lung, liver and spleen of IRF2$^{KO}$ compared to WT mice. However, these differences were only statistically significant in the liver ($p = 0.0042$), but not in the lung ($p = 0.3180$) or spleen ($p = 0.5515$) (Fig. 1B). Notably, the bacterial burden was over 100-fold greater in the liver, and tenfold greater in the lung and spleen in IRF2$^{KO}$ mice compared to WT mice. To understand the causes of the susceptibility of IRF2$^{KO}$ mice to *F. novicida*, we monitored the serum levels of a panel of cytokines using a Luminex assay, during infection (Fig. 1C). IFN-γ levels were reduced in the blood of IRF2$^{KO}$ mice, indicating that IRF2 plays a critical, albeit possibly indirect, role in regulating IFN-γ production. Surprisingly, IRF2$^{KO}$ mice displayed significantly higher serum IL-18 levels compared to WT mice. IRF2$^{KO}$ mice also exhibited increased serum levels of IL-6 (Fig. 1C), a cytokine potentially induced by IL-1β (Tosato and Jones, 1990), as well as increased IL-1β levels, which were significantly higher in IRF2$^{KO}$ mice than in WT mice (Fig. 1D). There were no major differences in IL-12 levels between IRF2$^{KO}$ and WT mice. However, a small decrease in IL-15 levels was observed in the IRF2$^{KO}$ mice compared to WT mice (Fig. 1C). Both IL-12 and IL-15 are key cytokines involved in regulating IFN-γ production (Koka et al, 2004). Importantly, no differences were observed in TNF levels, demonstrating that IRF2$^{KO}$ mice did not exhibit widespread cytokine dysregulation but had specific deficits in IFN-γ production. To confirm these results, we monitored serum IL-18 and IFN-γ levels using conventional ELISA. 48 h post-inoculation, IRF2$^{KO}$ animals showed a profound deficit in IFN-γ production (Fig. 1E), coupled with markedly elevated serum IL-18 levels compared to WT levels (Fig. 1F),

corroborating the Luminex results. IFN-β levels were significantly increased in IRF2$^{KO}$ mice compared to WT mice following *Francisella* infection (Fig. 1G).

Building on these results and on prior findings that murine IRF2 regulates GSDMD expression in vitro (Kayagaki et al, 2019), we investigated the role of IRF2 in GSDMD regulation in vivo. Our data indicate that IRF2 contributed to the regulation of GSDMD transcript and protein levels, both in naive (Fig. 1H,I) and infected mice (Fig. 1J,K), although the observed differences in *GSDMD* transcript levels did not reach statistical significance in the latter case. Of note, *GSDMD* transcript levels were induced (three- to four-fold increase depending on the organ) upon infection, with a statistically significant change observed only in the liver (Fig. EV1G). Caspase-1 (Fig. 1I) and ASC (Fig. 1K) levels were not substantially affected by IRF2 status. Interestingly, despite a large reduction in full-length GSDMD levels, the cleaved GSDMD fragment was specifically observed in the spleen from infected IRF2$^{KO}$ mice (Fig. 1K), possibly due to the higher bacterial burden.

Overall, our results show that IRF2$^{KO}$ mice are more susceptible to infection by *F. novicida* than control mice, which is associated with decreased expression of GSDMD and IFN-γ but increased levels of IL-18 and IL-1β.

## IRF2 regulates GSDMD level, inflammasome activation and IL-1β/IL-18 secretion in response to bacterial infections in bone marrow-derived macrophages

To better understand how IRF2 controls inflammasome and IL-1β/IL-18 production, we used bone marrow-derived macrophages (BMDMs) from IRF2$^{KO}$ and other mice and infected them with different bacteria. We initially infected them with *F. novicida*, which activates the AIM2 inflammasome (Jones et al, 2010; Fernandes-Alnemri et al, 2010). Endogenous protein expression was measured by capillary electrophoresis immunodetection. Full-length GSDMD (and to a further extent, cleaved GSDMD) was not detected in IRF2$^{KO}$ nor GSDMD$^{KO}$ BMDM (Fig. 2A). In contrast, BMDMs of all other genotypes expressed GSDMD at basal levels, its expression increasing following *F. novicida* infection. GSDMD cleavage was observed upon infection in WT and CASP11$^{KO}$ BMDMs, as expected, but was absent in AIM2$^{KO}$, ASC$^{KO}$, GSDMD$^{KO}$, and IRF2$^{KO}$ BMDMs. ASC levels followed the expected pattern and were not regulated by IRF2. Additionally, procaspase-1 (Pro-CASP1) levels were increased in IRF2$^{KO}$ BMDMs even in the absence of infection. Caspase-1 cleavage occurred in an AIM2- and ASC-dependent manner and was more pronounced in infected GSDMD$^{KO}$ and IRF2$^{KO}$ cells, likely due to the elevated proCaspase-1 levels combined with the absence of GSDMD-dependent release mechanisms. Indeed, cleaved caspase-1 was undetectable in the supernatant of both GSDMD$^{KO}$ and IRF2$^{KO}$ BMDMs.

To examine the functional outcomes of this regulation, we monitored cell death (Fig. 2B), and the secretion of IL-1β (Fig. 2C), IL-18 (Fig. 2D), and TNF (Fig. 2E) in BMDMs from all genotypes. At 6- and 8-h post-inoculation (p.i.), cell death and IL-1β secretion were fully abolished in *F. novicida*-infected IRF2$^{KO}$ and GSDMD$^{KO}$ BMDMs (Fig. 2B,C). As expected, AIM2 and ASC were also required for *F. novicida*-mediated cell death as well as IL-1β release. A minor contribution of caspase-11 in IL-1β and IL-18 was observed, especially at high MOI. This observation was unexpected due to the inability of *F. novicida* tetra-acylated LPS (Hagar et al,

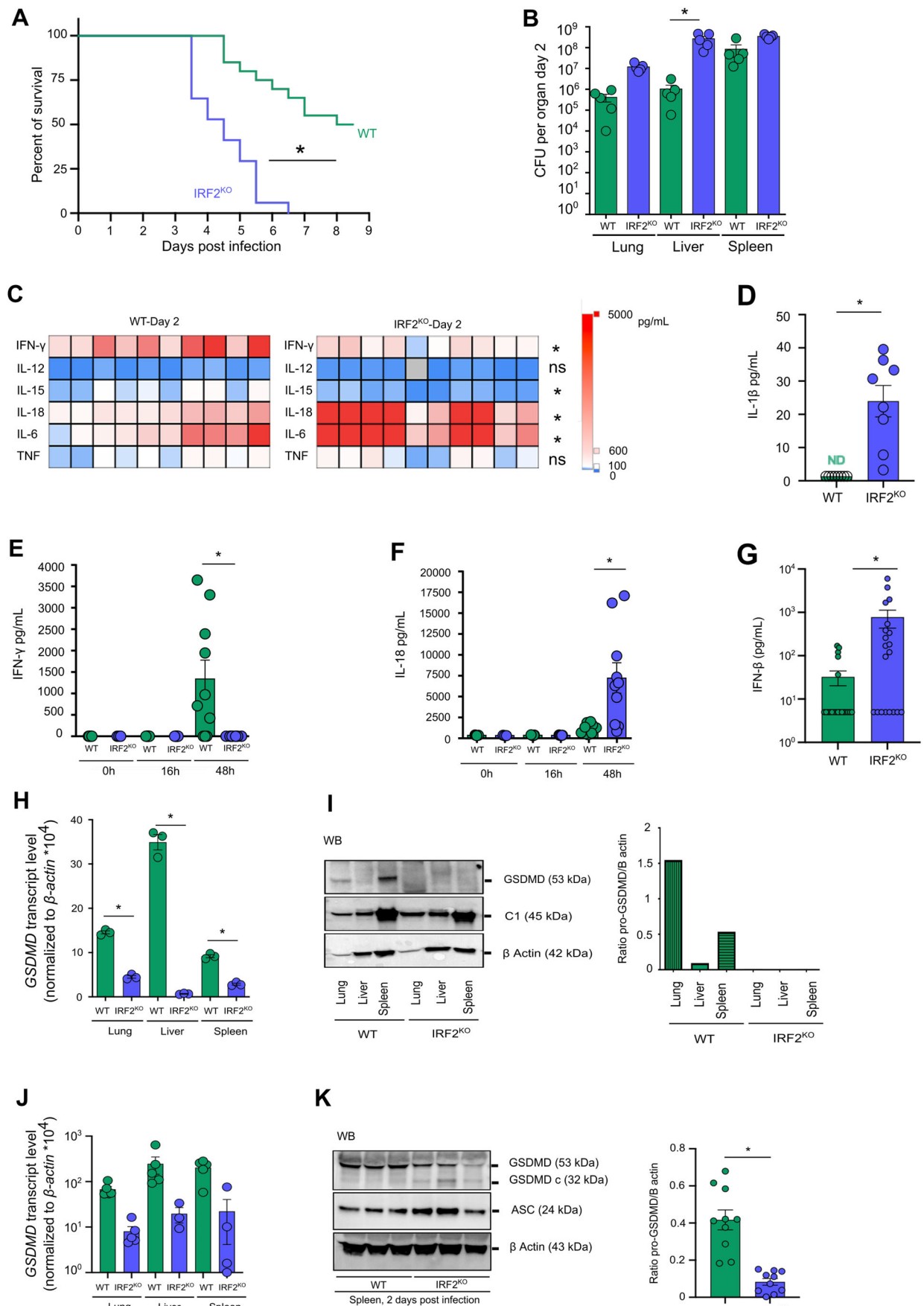

**Figure 1. IRF2<sup>KO</sup> are defective in GSDMD expression, IFN-γ production and highly susceptible to *F. novicida* infection.**

Mice of the indicated genotypes were subcutaneously (sc) injected with *F. novicida* or PBS. (A) Mice injected with $7 \times 10^2$ *F. novicida* were monitored for survival twice daily. (B) Mice injected with $2.5 \times 10^3$ *F. novicida* were assessed for bacterial burden in the lung, liver, and spleen at day 2 post-inoculation. (C, D) Serum cytokine levels were measured at day 2 post-inoculation by Luminex assay. (E, F) Serum cytokine levels were measured at day 2 post-inoculation by ELISA. (G) Serum cytokine levels were measured at day 2 post-inoculation by Luminex assay. (H) Naive mice injected with PBS were assessed for GSDMD transcript levels. (I) GSDMD, procaspase-1, and β-actin protein levels were assessed in naive mice. In (I), samples derive from the same experiment and gels/blots were processed in parallel. (J) GSDMD transcript levels were assessed in the spleen of mice injected with $2.5 \times 10^3$ *F. novicida* at 48 h post-inoculation. (K) Full-length and cleaved GSDMD (GSDMDc), ASC, and β-actin protein levels were assessed in the spleen at 48 h post-inoculation by Western blot. (A–K) Data represent one experiment (from three independent experiments) or represent pooled data from three independent experiments (A, C–G, J, K). Each dot corresponds to the value in one mouse, the bar shows the mean and the standard error of the mean (SEM). The data were collected as follows: (A) shows results from 15 WT and 17 IRF2<sup>KO</sup> mice, with data pooled from three independent experiments; (B) shows results from five WT and five IRF2<sup>KO</sup> mice; (C) shows results from ten WT and ten IRF2<sup>KO</sup> mice; (D) shows results from eight WT and eight IRF2<sup>KO</sup> mice; (E, F) show results from ten WT and ten IRF2<sup>KO</sup> mice; (G) shows results from 20 WT and 20 IRF2<sup>KO</sup> mice (detection threshold was set at 0.26 pg/mL per manufacturer's instructions; for mice falling below the detection threshold, the values appear as a line at the baseline of the bargraph); (H) shows data from three WT and three IRF2<sup>KO</sup> mice; (I) shows a representative blot from one of two independent experiments. (J) includes at least three WT and three IRF2<sup>KO</sup> mice. (K) The blot shows GSDMD expression from three WT and three IRF2<sup>KO</sup> mice. Log-rank Cox–Mantel test (A), Kruskal–Wallis analysis with Dunn's correction (B, E, J), unpaired *t*-test (C, D, G, K), and one-way ANOVA analysis with Sidak correction (F, H) were performed. * indicates $p < 0.05$; exact *p* values are listed below. P values : (A) <0.0001; (B) 0.0159; (C) IFN-γ: 0.0038; IL-12 : 0.2454; IL-15 : 0.0022; IL-18 : 0.0231; IL-6 : 0.0153; TNF: 0.3833; (D) 0.0003; (E) <0.0001; (F) <0.0001; (G) 0.0043; (H) Lung : <0.0001; Liver : <0.0001; Spleen : <0.0001; (K) <0.0001. Source data are available online for this figure.

2013; Lagrange et al, 2018) to activate caspase-11 and suggests that LPS molecules with higher acylation levels might be present at low frequency in *F. novicida*. IL-18 release was undetectable at 6 h p.i., and was produced under the same conditions as IL-1β at 8 h p.i. (Fig. 2D, left panel). Interestingly, at 24 h p.i., IL-18 levels increased in the supernatants of infected IRF2<sup>KO</sup> and GSDMD<sup>KO</sup> cells, albeit to a much lower extent than in WT macrophages, while IL-18 remained low to undetectable in AIM2<sup>KO</sup> and ASC<sup>KO</sup> BMDMs (Fig. 2D, right panel). At the same 24-hour time point, IL-1β release was still fully dependent on ASC, but measured at comparable levels in WT, IRF2<sup>KO</sup>, GSDMD<sup>KO</sup>, and CASP11<sup>KO</sup> cells (Fig. EV2A). These findings suggest that although both IL-1β and IL-18 are regulated by IRF2 and GSDMD, their detectable levels (secretion or degradation) follow distinct temporal patterns during *Francisella* infection. In particular, at late time points of infection, IL-1β levels appear to be less dependent on the IRF2-GSDMD axis than IL-18 release.

To further investigate the role of IRF2 in inflammasome activation, BMDMs were infected with *Escherichia coli* or exposed to outer membrane vesicles (OMVs), two stimuli known to activate the non-canonical caspase-11 inflammasome. IL-1β secretion was strongly reduced in BMDMs infected with *E. coli* or treated with OMVs in the absence of IRF2 (Fig. EV2B). However, the reduction did not reach the levels observed in GSDMD<sup>KO</sup> BMDMs, suggesting that type I IFN produced during the 24-h treatment period in response to *E. coli* LPS, may partially restore inflammasome activity, likely through the IRF2-independent induction of *GSDMD* expression. As expected, caspase-11 and ASC were required for IL-1β secretion in response to *E. coli* or OMV treatment, while AIM2 was not. TNF production was similar in all infected BMDM genotypes (Fig. 2E and Fig. EV2C), following infection with either *F. novicida, E. coli,* or OMV treatment. Some variability was observed between WT and IRF2<sup>KO</sup> samples, but these differences were inconsistent and lacked reproducibility.

Overall, these in vitro experiments demonstrate that IRF2 is an essential regulator of inflammasome functions through its direct effect on GSDMD protein levels. However, in vitro results with BMDMs do not fully align with the results obtained in vivo in the mouse model of tularemia, particularly regarding the high IL-18 levels observed in vivo but not in vitro.

## IRF2 controls IFN-γ production through a regulation at play in both infected cells and IFN-γ-producing cells

Next, we sought to understand the causes of the decreased IFN-γ production in IRF2<sup>KO</sup> mice infected with *F. novicida*. For this, we investigated the interplay between infected macrophages and IFN-γ-producing cells, using a previously established coculture system of infected BMDMs with splenic mononuclear cells (SMCs) (Pierini et al, 2013). Using this coculture system, *F. novicida* infection recapitulated IFN-γ production observed in vivo. As expected, no IFN-γ was produced either in the absence of BMDM infection or in the absence of SMCs (Fig. EV2D-MOI1; and Fig. EV2E-MOI10). To assess the impact of IRF2 in macrophages on IFN-γ production, WT, IRF2<sup>KO</sup> or GSDMD<sup>KO</sup> BMDMs were infected and co-cultured with WT SMCs (Figs. 3A-MOI1 and EV2F-MOI10). Both IRF2 and GSDMD in BMDMs were required to promote IFN-γ production from SMCs. Previous studies have shown that IFN-γ production by splenic mononuclear cells (SMCs) can be stimulated by combinations of cytokines such as IL-12, IL-15, and IL-18 (Nielsen et al, 2016; Otani et al, 1999). The addition of exogenous IL-18 fully complemented GSDMD<sup>KO</sup> macrophages and partially complemented IRF2<sup>KO</sup> macrophages. In contrast, supplementation with IL-15 alone had no effect on IFN-γ levels in both IRF2<sup>KO</sup> and GSDMD<sup>KO</sup> BMDM/WT SMCs co-cultures at the lower MOI. A significant increase in IFN-γ levels was observed only when IL-15 was combined with IL-18, indicating that IRF2 controls IFN-γ production by regulating GSDMD levels and the release of IL-18 in macrophages rather than through IL-15 signaling. Since IL-18 complementation was only partial in the IRF2<sup>KO</sup> BMDM/WT splenocytes coculture system, IRF2 may also regulate other BMDM functions required to promote optimal IFN-γ production by SMCs.

IL-18 was present in large quantities in the serum of infected IRF2<sup>KO</sup> mice at 48 h post infection. Yet, it did not result in IFN-γ production, suggesting that IRF2 may also play a role in the lymphoid compartment producing IFN-γ (Pierini et al, 2013). We thus conducted additional experiments, co-culturing WT BMDMs with SMCs from WT, IRF2<sup>KO</sup> or GSDMD<sup>KO</sup> mice (Figs. 3B-MOI1 and EV2G-MOI10). Even in the presence of infected WT BMDMs, SMCs from IRF2<sup>KO</sup> mice were unable to produce IFN-γ. In contrast, SMCs from GSDMD<sup>KO</sup> mice did not differ from WT splenocytes in

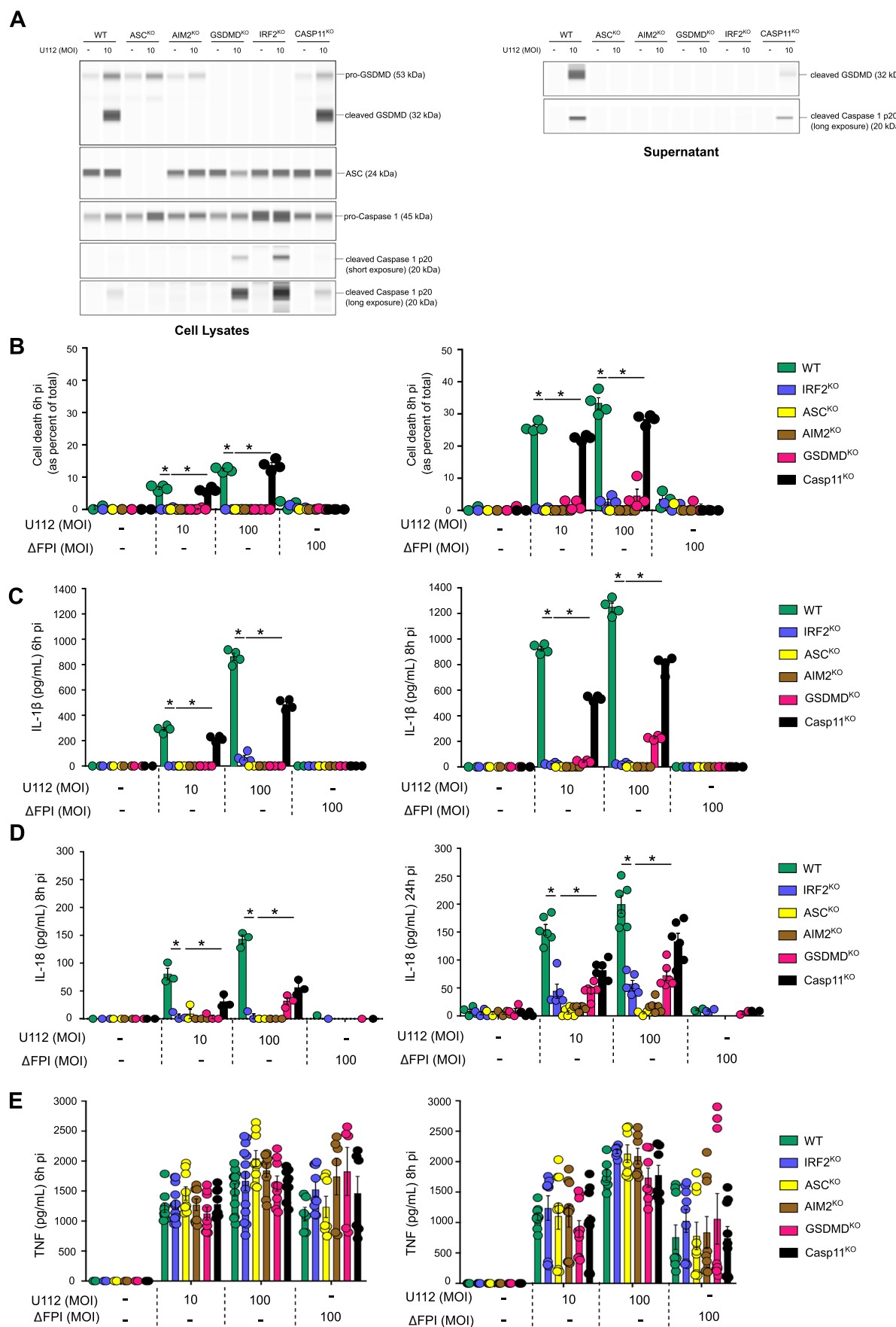

**Figure 2. IRF2^KO BMDMs are deficient in GSDMD expression and inflammasome activation in response to *F. novicida* infection.**

(A–E) BMDMs from the indicated genotypes were infected or not with *F. novicida* (U112) at an MOI of 10 or 100, or with its avirulent mutant ΔFPI. (A) Lysates (left panels) or supernatants (right panels) from BMDMs from the indicated genotypes infected or not with *F. novicida* at a MOI of 10 were analyzed by immunoblot 7 h p.i.. (B) Cell death was quantified by LDH assay and expressed as normalized values. (C) IL-1β, (D) IL-18, and (E) TNF concentrations were determined by ELISA. (B–E) Each dot corresponds to one replicate; the bar shows the mean and the standard error of the mean (SEM) of four replicates from one experiment representative of three independent experiments (B–D) or from the pooled data of three independent experiments (E). One-way ANOVA analysis with Dunnett correction (B–D) was performed. * indicates $p < 0.05$; exact $p$ values are listed below. $p$ values : (B) 6 h p.i., from left to right: <0.0001, <0.0001, <0.0001, <0.0001; 8 h p.i., from left to right: <0.0001, <0.0001, <0.0001, <0.0001; (C) 6 h p.i., from left to right: <0.0001, <0.0001, <0.0001, <0.0001; 8 h p.i., from left to right: <0.0001, <0.0001, <0.0001, <0.0001; (D) 8 h p.i., from left to right: <0.0001, 0.0351, <0.0001, <0.0001; 24 h p.i., from left to right: <0.0001, 0.0366, <0.0001, 0,0027. Source data are available online for this figure.

their ability to produce IFN-γ, strongly suggesting a functional defect of IFN-γ-producing lymphocytes in IRF2^KO mice. Supplementation with IL-15 or IL-18 did not complement the IFN-γ defect in IRF2^KO splenocytes, indicating that, in the SMCs compartment, IRF2 controls IFN-γ production independently of GSDMD, IL-15 and IL-18.

## IRF2 regulates NK cell peripheral cell number and maturation at steady state and during *Francisella novicida* infection

Our data suggest a role for IRF2 in IFN-γ production during infection. IFN-γ can be produced by multiple innate and adaptive lymphocyte subsets including Natural Killer (NK), NKT, CD8, and CD4 T cells. Since previous articles reported defective development of NK cells in other mouse models of IRF2 deficiency, we conducted a detailed flow cytometry analysis of immune cell composition in WT and IRF2^KO mice infected with *F. novicida*.

A significant decrease in the frequency of NK cells was observed in the blood of infected IRF2^KO, compared to WT mice (Fig. 4A). This result was confirmed by t-distributed stochastic neighbor embedding (t-SNE) analyses performed on lung, liver and spleen cell suspensions (Fig. 4B). NK cell numbers were strongly reduced in IRF2^KO mice compared to WT mice in all organs analyzed (Figs. 4B,C and EV3A–C).

NK cells undergo a maturation process characterized by the expression of surface markers such as CD11b and CD27, during which they acquire their effector functions (Degouve et al, 2016). Three distinct maturation stages have been defined, progressing from the most immature to the mature state, as follows: (i) CD11b^low CD27^high, (ii) Double Positive (DP), and (iii) CD11b^high CD27^low (Chiossone et al, 2009). In *IRF2* deficient mice, not only NK cell numbers were reduced in the periphery, but they also exhibited impaired maturation (Fig. 4D). Specifically, in IRF2^KO mice, a significant proportion (60 to 80%) of NK cells in the lung, liver and spleen, exhibited an immature phenotype (Fig. 4D). By contrast, NK cells in WT mice predominantly exhibited mature phenotypes (either DP or CD11b^high CD27^low).

Next, we investigated whether the reduction in NK cell numbers was also observed in naive mice. As shown in Fig. 4E,F, a similar reduction in NK cell numbers was observed in IRF2^KO mice prior to infection, whereas innate lymphoid cell (ILC) subsets were not statistically different compared to WT mice (Fig. 4E,F). In particular, ILC1, ILC2, and ILC3 were not significantly impacted by the absence of *IRF2*, highlighting the selective role of IRF2 in regulating NK cell development. Interestingly, the number of NK cells in the bone marrow of IRF2^KO mice was slightly increased

compared to WT mice (Fig. EV3D), suggesting that loss of IRF2 may slow the exit of NK cells from the bone marrow. However, bone marrow NK cells from IRF2^KO mice displayed a strong deficit in the expression of antibacterial effectors, including Granzyme A (GzmA) and IL-18 receptor (IL-18R), in this organ (Fig. EV3E).

Furthermore, the maturation defect in peripheral IRF2^KO NK cells was also associated with reduced expression of GzmA and IL-18R, although not all differences were statistically significant (Figs. 4G,H and EV3F,G), suggesting compromised cytotoxic activity and decreased responsiveness to IL-18, respectively.

T-box expressed in T cells (Tbet) and Eomesodermin (Eomes) are two critical transcription factors in NK cell maturation and functions (Daussy et al, 2014; Zhang et al, 2018). Eomes acts upstream of Tbet within the transcriptional network that governs NK cell development. In IRF2^KO naive mice, Eomes expression in NK cells was slightly reduced in the lung compared to their WT counterparts (Fig. 4I). Additionally, both Eomes and Tbet expression levels were lower in NK cells in the spleen of IRF2^KO naive mice, compared to WT mice, with a statistically significant reduction observed only for Eomes (Fig. 4J).

These findings suggest that IRF2 plays a role in promoting NK cell maturation, potentially acting upstream of Eomes. Both Tbet and Eomes are essential for the establishment and homeostasis of the NK cell population, which likely explains the reduced NK cell numbers in IRF2^KO mice, and the resulting dramatic decline in IFN-γ production.

Finally, to assess whether IRF2 plays a cell-intrinsic role in NK cell development, we conducted bone marrow chimera experiments. Irradiated CD45.1 recipient mice were reconstituted with two types of donor bone marrow cells: (i) WT CD45.1/CD45.2 cells and (ii) IRF2^KO CD45.2/CD45.2 cells. Donor cells were mixed at either a 50:50 ratio (50% WT/50% IRF2^KO), or a 100:0 ratio (100% WT) as a control. In mixed bone marrow chimera mice (50% WT/ 50% IRF2^KO), very few NK cells derived from IRF2^KO progenitors (Fig. 4K) and those cells showed a pronounced immature phenotype (Fig. 4L). In contrast, mature and terminally differentiated NK cells developed from WT progenitors at similar frequencies in WT mice, simple bone marrow chimera mice (100% WT) and mixed bone marrow chimera mice (50% WT/50% IRF2^KO) (Fig. 4K). These findings demonstrate an intrinsic role of IRF2 in NK cells, which failed to mature in the mixed chimera, even though this environment supported proper NK cell maturation and development, as evidenced by the normal development of engrafted WT NK cells (Fig. 4L).

Overall, these findings underscore the essential role of IRF2 in orchestrating NK cell development, survival, and maturation, in the context of *F. novicida* infection.

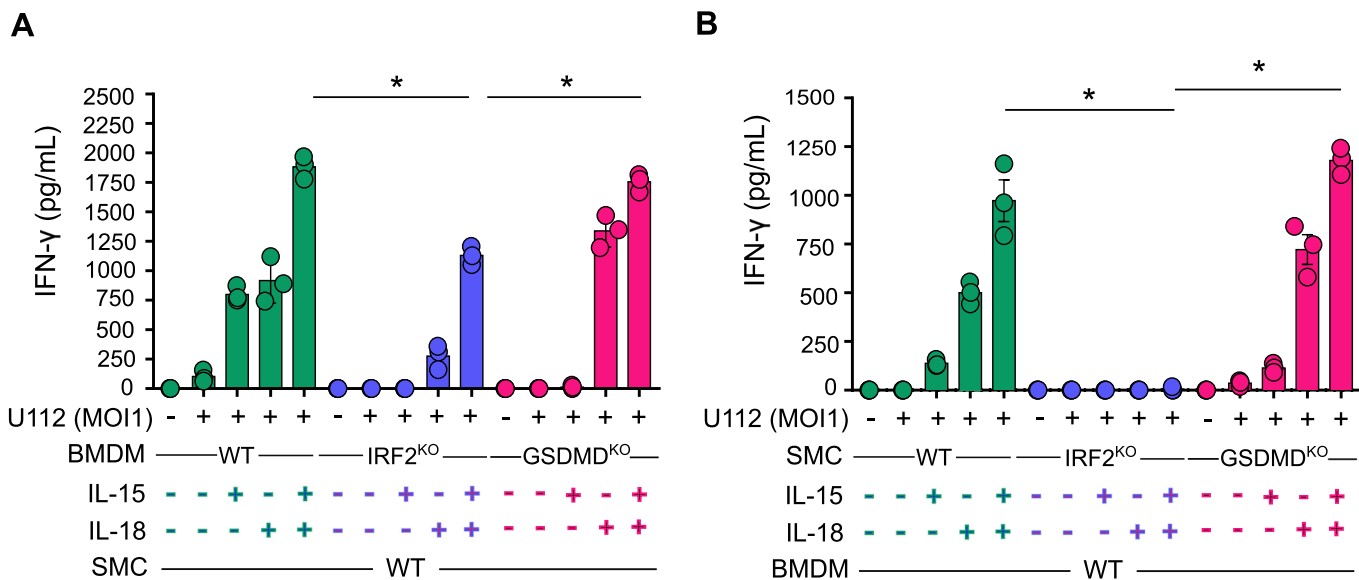

**Figure 3.  IRF2 controls IFN-γ production through a dual regulation in infected cells and IFN-γ-producing cells.**

(A) BMDMs from the indicated genotypes were infected or not with *F. novicida* at a MOI of 1, and then co-cultured with WT splenic mononuclear cells (SMCs) in the presence or not of IL-15/IL-15R complexes (10 ng/mL), IL-18 (200 ng/mL), or both IL-15/IL-15R complexes and IL-18. (B) WT BMDMs were infected or not with *F. novicida* at a MOI of 1, and then co-cultured with splenic mononuclear cells (SMCs) of the indicated genotypes in the presence or not of IL-15/IL-15R complexes (10 ng/mL), IL-18 (200 ng/mL), or both IL-15/IL-15R complexes and IL-18. Supernatants were collected 24 h later, and the level of IFN-γ cytokine was measured by ELISA. One dot represents the value from one replicate, the bar represents the mean and the standard error of the mean (SEM) of three replicates from one experiment, representative of three independent experiments. One-way ANOVA analysis with Dunnett correction (A, B) was performed. * indicates $p < 0.05$; exact $p$ values are listed below. $P$ values : (A) from left to right: <0.0001, <0.0001; (B) from left to right: 0.0001, <0.0001. Source data are available online for this figure.

## GSDMD[KO] mice are highly susceptible to *Francisella novicida* infection but do not recapitulate the IRF2[KO] phenotype

The above findings suggest that IRF2[KO] mice display both GSDMD-dependent and GSDMD-independent immune defects. To clarify the contribution of each pathway, IRF2[KO] and GSDMD[KO] were compared side-by-side following *F. novicida* inoculation. GSDMD[KO] and IRF2[KO] mice were equally sensitive to *F. novicida* infection (Fig. 5A), with mice from both genotypes showing higher CFU counts in lung, liver and spleen, compared to WT mice (Fig. 5B). Interestingly, both IRF2[KO] and GSDMD[KO] mice had higher serum levels of IL-18 than WT mice upon infection with *F. novicida* (Fig. 5C). However, the behavior of both genotypes diverged at the level of IFN-γ production. Indeed, compared to WT mice, IRF2[KO] mice produced less IFN-γ, whereas GSDMD[KO] mice exhibited increased IFN-γ production (Fig. 5D). The increased IL-18 levels observed in both IRF2[KO] and GSDMD[KO] mice may be due to the increased bacterial burden observed in these two mice (Fig. 5B). To avoid this bacterial burden-associated bias, we injected mice with LPS at 3 µg per gram of body weight for 6 h. IL-18 levels were reduced in both IRF2[KO] and GSDMD[KO] mice compared to WT mice (Fig. 5E). These results demonstrate that in vivo, in the absence of confounding factors, IRF2 (and GSDMD) positively controls IL-18 levels.

Given that GSDMD is expressed in NK cells throughout their differentiation process (Fig. EV4A), we wondered if the NK cell deficiency in IRF2[KO] mice could be related to *GSDMD* regulation. NK cell maturation and function were examined in GSDMD[KO]

mice, comparing them to WT and IRF2[KO] mice. Both the number and maturation profile of NK cells were comparable between GSDMD[KO] mice and WT mice (Fig. 5F–G). We then examined whether this pattern held in naive conditions, which was indeed the case, as shown in Fig. 5H. Furthermore, NK cells from both WT and GSDMD[KO] naive mice expressed the two pivotal activation markers, Granzyme A (GzmA) (Fig. 5I) and IL-18 receptor (IL-18R) (Fig. 5J), whereas NK cells in IRF2[KO] mice lacked expression of these markers. NK cell proliferation, as assessed by the Ki67 assay, was significantly higher in IRF2[KO] mice, compared to both WT and GSDMD[KO] mice, indicating increased turnover (Fig. 5K).

These findings suggest that while GSDMD[KO] and IRF2[KO] mice are both highly susceptible to *F. novicida* infection, there are notable differences in cytokine production and NK cell functions between the two knockout mouse models. Specifically, GSDMD[KO] mice exhibit altered cytokine production but retain normal NK cell maturation and function, contrasting with the broader defects in IRF2[KO] mice.

## IFN-γ therapy, and not lymphokine-activated killer (LAK) cell therapy, prolongs the survival of infected IRF2[KO] mice

Given the NK cell deficiency in *IRF2*[KO] mice, we tried to rescue the NK cell compartment by adoptive transfer of lymphokine-activated killers (LAK). LAK cells are primary splenic NK cells amplified ex vivo with IL-2 to boost their effector functions. The transfer of LAK cells led to a notable increase in NK cells in IRF2[KO] mice (Fig. EV5A). Yet, the transfer did not improve the resistance of IRF2[KO]

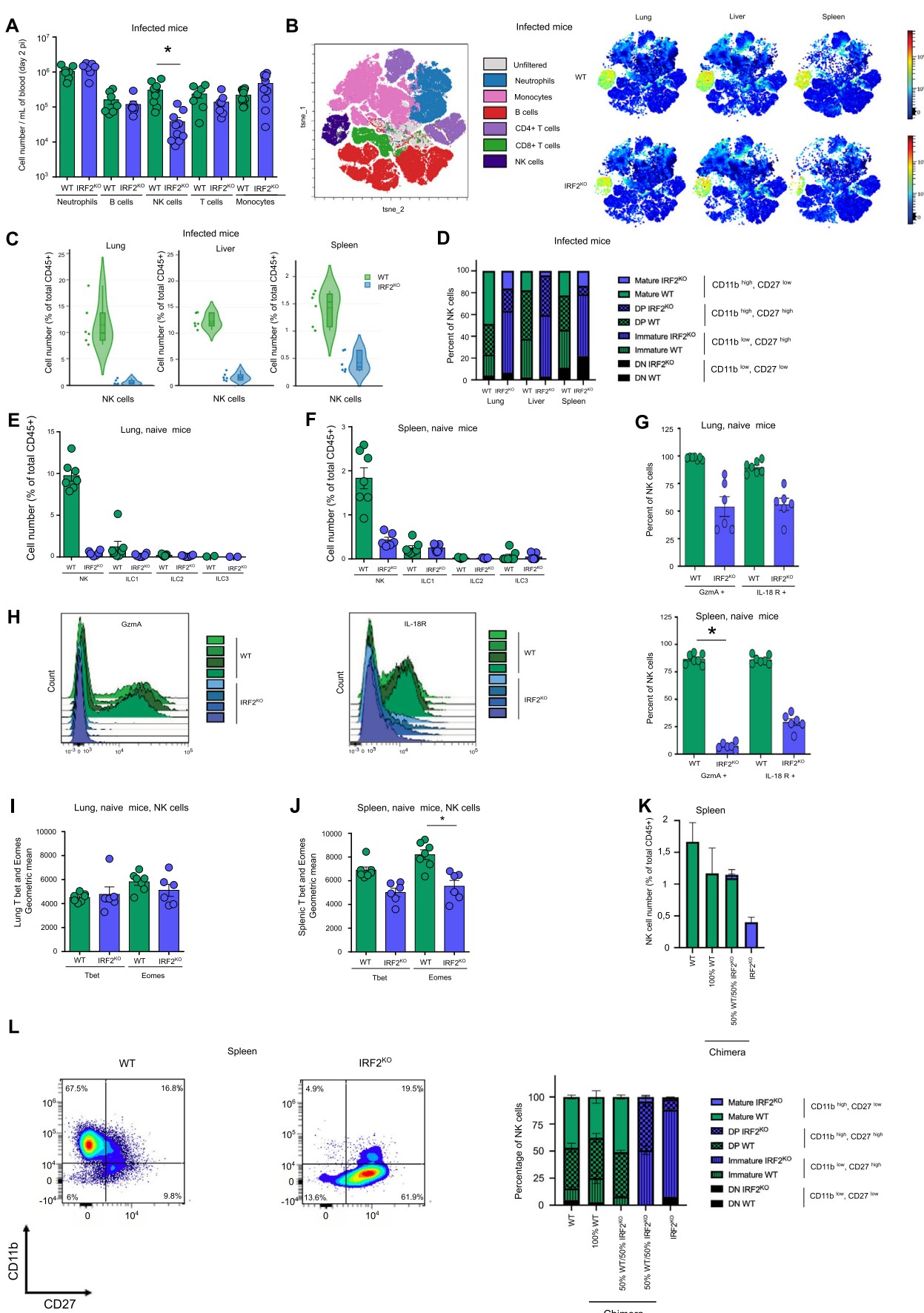

◀ **Figure 4. IRF2^KO NK cells have an intrinsic defect in maturation and effector functions acquisition.**

(A–D) WT or IRF2^KO mice were infected with 2500 CFU for 2 days. (A) Absolute numbers of the indicated CD45+ subsets in infected mouse blood are shown (seven mice per group). (B, left panel) A t-SNE representation of immune cell populations from wild-type (WT) and IRF2^KO mice, with cells color-coded by subsets as indicated, is displayed. (B, right panel) t-SNE heatmaps showing the NK cell marker NK1.1 expression in lung, liver and spleen in three WT (top) and three IRF2^KO (bottom) mice. (C) Violin plots comparing the percentage of NK cells relative to the total CD45+ cell population in lung, liver and spleen between six WT and six IRF2^KO mice are shown. The embedded box plots indicate the median (central line), the 25^th and 75^th percentiles (box), and the minimum and maximum values (whiskers). Each dot represents an individual mouse. (D) Quantification of each maturation subset is shown; the bar shows the mean of the values from four mice per group. (E–J) Results from seven WT or six IRF2^KO naive mice are shown. (E, F) Frequency of CD45+ innate lymphoid subsets in the lung (E) and in the spleen (F) are displayed. (G, H) Frequency of GzmA+, IL-18R+, or Ki67+ cells among total NK cells in the lung (G) and spleen (H) are shown. (H) GzmA and IL18R expression levels in four wild-type (WT, green) and 4 IRF2^KO (blue) mice are displayed using overlaid histograms. (I, J) Tbet and Eomes geometric mean are shown in the lung (I) and spleen (J). (K, L) Bone marrow chimera experiment was performed using four WT and four IRF2^KO mice. The frequency of NK cells among total splenic CD45+ cells (K) and the phenotypic analysis of their maturation based on CD11b and CD27 expression in the spleen (L) were assessed in two types of recipient mice, either reconstituted with 100% of WT donor cells (100% WT) or with a mixture of 50% of WT donor cells and 50% of IRF2^KO donor cells. The dot plot highlights maturation of NK cells derived from WT (left panel) or IRF2^KO (right panel) progenitors, in a 50/50 chimera, with data concatenated from four mice. (L) Quantification of each maturation subset is shown; the bar shows the mean with n = 4 mice per group. (A, E–J) Each symbol represents the value of an individual mouse. (A, E–L) Bars indicate the mean with SEM. Kruskal–Wallis analysis with Dunn's correction was performed. Two tailed p values are shown with the following nomenclature *p < 0.05. * indicates p < 0.05; exact p values are listed below. P values: (A) 0.0223, (H) 0.236, (J) 0.0264. Source data are available online for this figure.

or GSDMD^KO mice to *F. novicida* infection (Fig. EV5B,D,E). Counterintuitively, the transfer even led to increased susceptibility to *F. novicida* infection in WT mice (Fig. EV5B,C).

Subsequently, we attempted to rescue these mice by injecting recombinant IFN-γ (rIFN-γ) daily for the first 5 days of infection. rIFN-γ therapy prolonged the survival of infected IRF2^KO and GSDMD^KO mice, and even rescued 75% of the infected GSDMD^KO mice (Fig. 6A–D). To elucidate the mechanism of action of IFN-γ in IRF2^KO mice, we assessed whether IFN-γ restored NK cell frequency and GSDMD transcript and protein levels in IRF2^KO mice. rIFN-γ treatment led to an increase in the frequency of NK cells in both the blood (Fig. 6E) and spleen (Fig. 6H) of naive mice, regardless of the presence or absence of IRF2. Interestingly, despite this expansion, the cells maintained an immature phenotype, as shown in Fig. 6F (blood) and Fig. 6I (spleen). Notably, granzyme A expression showed a trend toward upregulation, as shown in Fig. 6G (blood) and Fig. 6J (spleen), suggesting that, in the presence of IFN-γ, IRF2-deficient NK cells may retain some cytotoxic potential despite their immature status. Similarly, IFN-γ treatment resulted in increased GSDMD transcript (Fig. 6K) and protein (Fig. 6L) levels in the spleen of both WT and IRF2^KO mice. These results indicate that IFN-γ acts independently of IRF2 to promote the expansion or mobilization of NK cells in the periphery and to up-regulate GSDMD expression levels and restore immune functions in IRF2^KO mice.

## Discussion

IRF2 has emerged as a key regulator of inflammasomes, regulating caspase-4 and GSDMD expression levels in human cells and GSDMD levels in BMDMs (Kayagaki et al, 2019; Benaoudia et al, 2019; Thygesen and Stacey, 2019). Importantly, gasdermin levels regulate the switch between pyroptosis and apoptosis, and the expression levels of several gasdermin have an impact on disease (e.g., asthma and cancers) and treatment (e.g., chemotherapy) (Zhou et al, 2020; Bourdonnay and Henry, 2022). Yet, whether GSDMD transcriptional regulation has an in vivo impact on disease and notably on the progression of bacterial infections, remains unknown. In this context, the role of IRF2 in vivo remained to be characterized.

## IRF2 regulates GSDMD level and inflammasome activation in response to *F. novicida*

Our study demonstrates a crucial role of IRF2 in regulating immune responses against *F. novicida*, that is at least partially mediated by its regulatory function on GSDMD. Indeed, reduced levels of GSDMD transcript and protein were observed in multiple organs in IRF2^KO mice during *F. novicida* infection compared to WT mice. This downregulation of GSDMD, was associated with increased susceptibility to infection, as evidenced by elevated bacterial counts in the organ targeted by *Francisella*. The largest difference in bacterial counts between WT and IRF2^KO was observed in the liver (Fig. 1B), where IRF2 invalidation has the strongest effect on GSDMD transcript levels (Fig. 1H). We and others have shown that IRF1 can complement IRF2 deficiency for inflammasome gene regulation (Benaoudia et al, 2019; Kayagaki et al, 2019). IRF1 levels vary at steady state between different organs and are lower in the liver compared to the spleen (Fig. EV4B), possibly explaining the organ-specific magnitude of IRF2 effects.

GSDMD is known to play a crucial role in the immune response to *F. novicida* (Zhu et al, 2018), notably through the regulation of IL-18 levels and of the IL-18/IFN-γ cascade. Interestingly, this decrease in IFN-γ contrasts with IFN-β levels, which were increased in IRF2^KO mice relative to WT mice following *Francisella* infection. Importantly, since IFN-β is a positive regulator of the *F. novicida* inflammasome (Henry et al, 2007), the impaired inflammasome activation observed in these animals cannot be attributed to the deregulated type I interferon responses. Surprisingly, IL-18 levels were significantly increased in IRF2^KO compared to WT mice at 48 h post-inoculation. Elevated levels of IL-18 were also observed in the serum of GSDMD^KO mice compared to WT mice. This increase in IL-18 levels could be due to the maintenance of living cells with a sustained level of caspase-1 activity in the cytosol (Fig. 2A), allowing a prolonged cleavage of pro-IL-18 into its mature form. Supporting this GSDMD-independent role of caspase-1, IL-18 levels remain low in *F. novicida*-infected ASC^KO and Caspase-1/11^KO mice (Pierini et al, 2013). Alternative secretion mechanisms for IL-18, independent of the GSDMD conduit/-mediated pyroptosis, may facilitate the release of IL-18. These mechanisms remain to be investigated but may include the unconventional

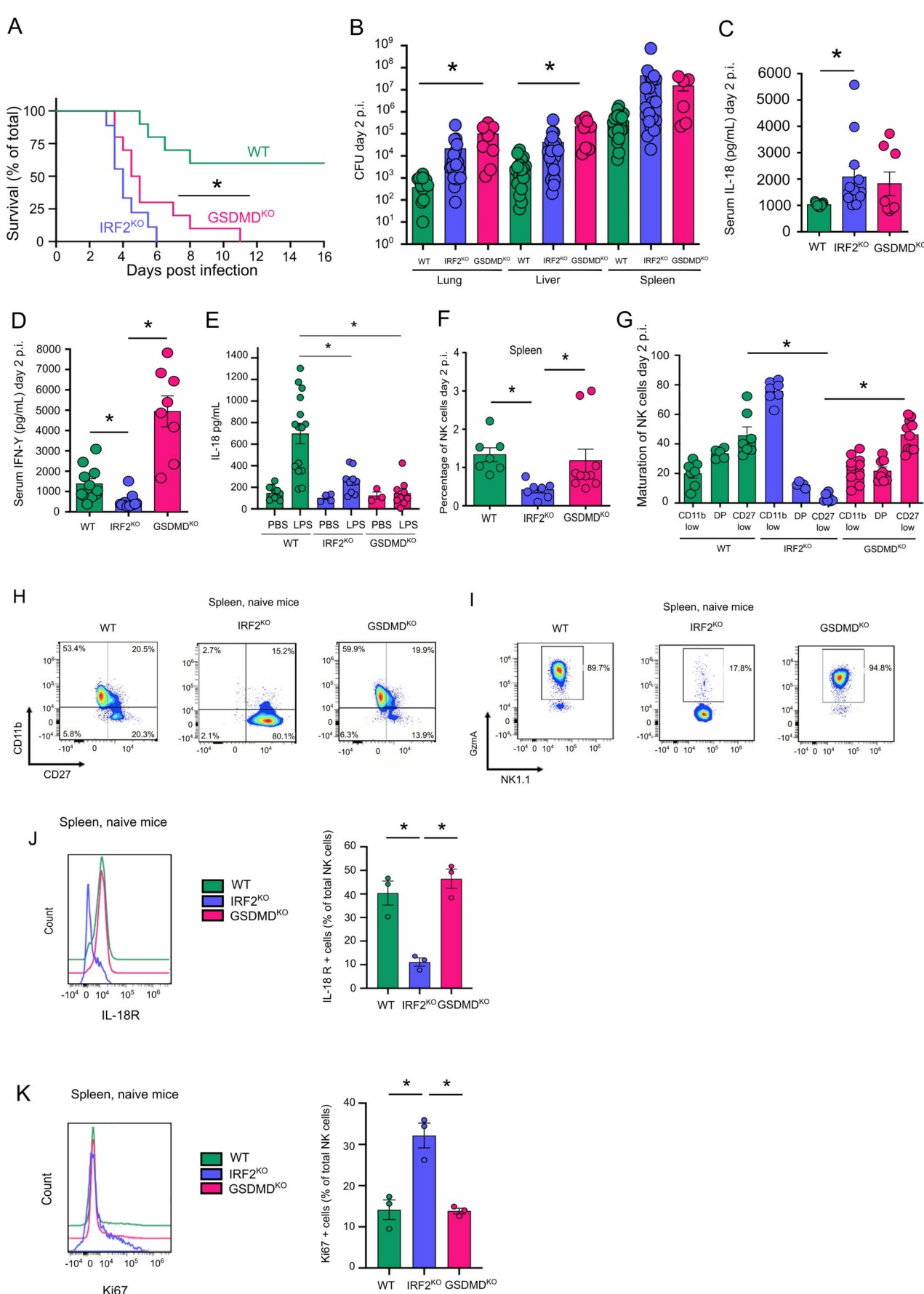

**Figure 5. GSDMD$^{KO}$ mice are highly susceptible to *F. novicida* infection but do not recapitulate IRF2$^{KO}$ immune defects.**

(A–G) Mice of the indicated genotypes were subcutaneously (s.c.) injected with 1000 CFU (A) or 2500 CFU (B–D, F, G), or intraperitoneally (i.p.) injected with 3 µg of LPS per gram of body weight per mouse (E). (H–K) Mice of the indicated genotypes were s.c. injected with PBS. (A) Survival was monitored twice a day using ten WT mice, nine IRF2$^{KO}$ mice, and ten GSDMD$^{KO}$ mice, with data pooled from two independent experiments. (B–G) Bacterial burden in the lung, liver, spleen (B); IL-18 (C, E), and IFN-γ (D) concentrations in the serum are shown. (F) Frequency of NK cells among total CD45$^+$ cells and (G) maturation of NK cells using CD11b and CD27 markers are shown. The data were collected as follows: (B) shows results from 21 WT, 22 IRF2$^{KO}$, and 8 GSDMD$^{KO}$ mice; (C, D) show results from 11 WT, 12 IRF2$^{KO}$, and 8 GSDMD$^{KO}$ mice; (E) shows results from 11 WT + PBS, 15 WT + LPS, 4 IRF2 + PBS, 9 IRF2 + LPS, 3 GSDMD + PBS, and 9 GSDMD + LPS mice. (F, G) show results from seven WT, seven IRF2$^{KO}$, and ten GSDMD$^{KO}$ mice. (H–K) WT, IRF2$^{KO}$, and GSDMD$^{KO}$ naive mice were used. (H) The maturation of NK cells using CD11b and CD27 markers are shown. (I) GzmA expression, (J) IL-18-R expression, and (K) Ki67 in NK cells were assessed by flow cytometry. (B–G, J, K) Each symbol represents the value of an individual mouse; the bars indicate the mean with SEM. (H, I) Flow cytometry pseudo-color dot plots comparing maturation stage (H) or GzmA expression (I) in spleen from WT, IRF2$^{KO}$, or GSDMD$^{KO}$ mice. (H, I) These flow cytometry plots from one mouse are representative of results obtained from three mice. (J, K) Overlaid histograms showing NK cell phenotype in WT (green), IRF2$^{KO}$ (blue), or GSDMD$^{KO}$ (pink) mice. Each curve represents data concatenated from three mice. Corresponding NK cell frequencies are displayed in the bar graphs on the right. (A) Log-rank Cox–Mantel test, (B–D, F) Kruskal–Wallis analysis with Dunn's correction, (E, G, J, K) one-way ANOVA analysis with Tukey correction were performed. * indicates $p < 0.05$; exact $p$ values are listed below. $p$ values: (A) WT vs IRF2$^{KO}$: <0.0001 and WT vs GSDMD$^{KO}$: 0.0014; (B) from left to right: 0.0155, 0.0223; (C) 0.0113; (D) from left to right: 0.0433, <0.0001; (E) from left to right: 0.0002, <0.0001; (F) from left to right: 0.0010, 0.0335; (G) from left to right: <0.0001, <0.0001; (J) from left to right: 0.0044, 0.0017; (K) from left to right: 0.0031 and 0.0028. Source data are available online for this figure.

secretion of membrane-associated IL-18 (Monteleone et al, 2018; Bellora et al, 2012), the exocytosis of IL-18 -containing vesicular carriers (Gardella et al, 2000; Li and Jiang, 2023), a potential partial complementation by other gasdermin family members or a passive release through GSDMD-independent cell death and membrane permeabilization (Broz, 2023). In vitro in infected BMDMs, IL-18 secretion was observed at 24 h p.i. in both IRF2$^{KO}$ and GSDMD$^{KO}$ BMDM. Yet, this secretion remains much lower than the secretion observed in WT BMDMs. Additional regulatory mechanisms or environmental cues present in the in vivo setting may modulate cytokine secretion differently than what is observed in isolated BMDM cultures. Alternatively, the higher bacterial counts observed in the organs of IRF2$^{KO}$ and GSDMD$^{KO}$ mice ( ~ 100-fold) compared to WT mice may account for this IL-18 increase. In agreement with this latter hypothesis, IL-18 levels were reduced in both IRF2$^{KO}$ and GSDMD$^{KO}$ mice compared to WT mice in a LPS challenge model (Fig. 5E).

Besides its well-established function in mediating cytokine release and pyroptotic cell death, GSDMD may have a direct antibacterial action through the binding of cardiolipin, a lipid found in bacterial membranes (Liu et al, 2016). Although *F. novicida* has an atypical cell envelope, its genome encodes the cardiolipin synthetase *ybhO* and may thus be directly sensitive to the bacteriolytic activity of GSDMD N-terminal fragment.

This study strengthens the role of IRF2 as a positive regulator of GSDMD expression. Interestingly, elevated basal procaspase-1 mRNA (Cuesta et al, 2007) and protein levels (Fig. 2A) were observed in IRF2$^{KO}$ macrophages, supporting a repressive role of IRF2 on caspase-1 transcription. Based on the position of GSDMD downstream of caspase-1 and in vitro data, IRF2 appears as a positive regulator of the inflammasome, although we cannot exclude that IRF2 may have context-dependent roles, where it may predominantly act either as an activator or a repressor of inflammasome functionality.

## IRF2 plays a crucial role in regulating NK cell development, maturation and therefore, IFN-γ production

IRF2 acts as a checkpoint for NK cell maturation, controlling the last steps of their maturation in the bone marrow (Taki et al, 2005; Lohoff et al, 2000; Li et al, 2016). Accordingly, we observed decreased levels of mature NK cells in the periphery of IRF2$^{KO}$ mice and a large increase in immature NK cells in the bone marrow. The immature state of IRF2$^{KO}$ NK cells, as defined by the classical CD11b/CD27 markers, is also evident when considering Eomes and Tbet levels, two transcription factors sequentially promoting the maturation of NK cells (Zhang et al, 2021). Our results thus position the IRF2 checkpoint in NK cells upstream of the Eomes/Tbet cascade. Importantly, in the context of bacterial infections, our study demonstrates that IRF2$^{KO}$ NK cells present reduced levels of the key antibacterial NK cell effectors, GzmA and IL-18R. The reduction in IL-18R coupled to the reduction in NK cell number is likely key to explain the inability of IRF2$^{KO}$ mice to mount an IFN-γ response and effective anti-*F. novicida* responses.

This double deficit in NK cell number and functionality led us to test whether NK cell therapy could rescue IRF2 deficiency and promote resistance to *F. novicida* infection. We used LAK with the assumption that it would allow a maximal cytokine production in infected mice. Yet, we did not observe a significant improvement in the survival of IRF2$^{KO}$ mice upon LAK cell transfer. Several factors may contribute to the ineffectiveness of NK cell therapy during *F. novicida* infection. First, the ability of IL-18 to activate NK cell functions may be counteracted by other immune factors present in infected mice, such as prostaglandin E$_2$ (PGE$_2$) or IL-6, that may inhibit NK cell cytotoxicity (Bonavita et al, 2020; Cifaldi et al, 2015). Second, LAK cells may be unable to migrate to the key microenvironment to establish the proper spatiotemporal process required to mount an efficient antibacterial immune response.

Not only did LAK cells fail to rescue IRF2$^{KO}$ mice, but their transfer into WT mice also resulted in an increased susceptibility to *F. novicida* infection, suggesting that in this context, LAK cells may be detrimental to the innate immune response against *F. novicida*. Adoptive transfer of ILC2s was also found to be detrimental to the control of *Francisella tularensis* infection, exacerbating the severity of infection (Dow et al, 2023). LAK cells may have a detrimental effect during infection with intracellular bacteria due to their secretion of interleukin-10 (IL-10) (Clark et al, 2020). Whether the detrimental effects of LAK cells and ILC2 adoptive transfers are mediated by the same mechanism (including in part via the production of the T helper type (Th)2 cytokine IL-5 (Dow et al, 2023)) remains to be deciphered.

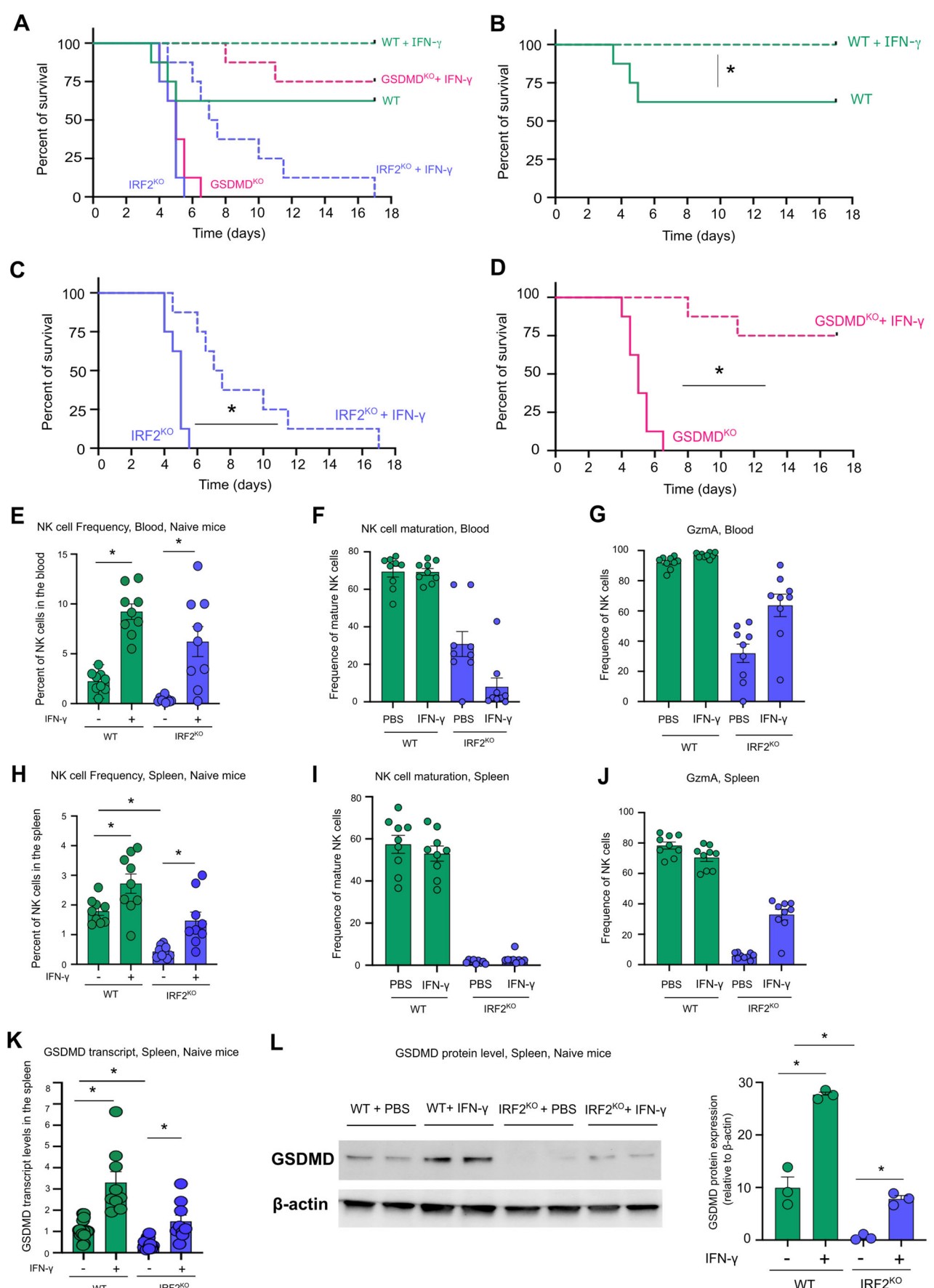

**Figure 6. Survival of IFN-γ-treated mice and effect of IFN-γ on NK cell frequency and GSDMD expression.**

(A–L) Mice of the indicated genotypes were subcutaneously inoculated with 1000 CFU (A–D) or with PBS (E–L), with or without recombinant murine IFN-γ, for 5 days starting the day of the infection. (A–D) Survival was monitored twice a day (8 mice were used per group, except in the WT + IFN-γ group, where ten mice were included, with data pooled from two independent experiments). (E–J) NK cell frequency (E, H), maturation (F, I), and GzmA expression (G, J) in the blood (E–G) and spleen (H–J) (results from nine mice per group are shown). (K, L) GSDMD transcript (K) and protein (L) levels were monitored in the spleen. (K) Results are shown from 17 mice in the WT + PBS group; 9 mice in the WT + IFN-γ group; 19 mice in the IRF2^KO + PBS group, and 9 mice in the IRF2^KO + IFN-γ group. (L) The blot represents samples from two individual mice, and the bar graph data correspond to three mice per group. Each symbol represents the value of an individual mouse; the bars indicate the mean with SEM. Log-rank Cox–Mantel test (A–D) and one-way ANOVA analysis with Tukey correction (E, H, K, L) were performed.* indicates $p < 0.05$; exact $p$ values are listed below. $p$ values: (B) WT vs WT + IFN-γ: 0.0376; (C) IRF2^KO vs IRF2^KO + IFN-γ: 0.0010; (D) GSDMD^KO vs GSDMD^KO + IFN-γ: <0.0001; (E) from left to right: <0.0001, 0.0002; (H) from left to right: 0.0374, 0.0164; (K) from left to right: <0.0001, 0.0477, 0.0061; (L) from left to right: <0.0001, 0.0016, 0.0077. Source data are available online for this figure.

The direct comparison of GSDMD^KO and IRF2^KO mice highlighted that IRF2 regulates in a GSDMD-independent manner, the level of the IFN-γ, a key cytokine orchestrating immune responses against *F. novicida* (Wallet et al, 2017). Given the failure of LAK therapy, we thus evaluated the ability of rIFN-γ to rescue innate immune responses in IRF2^KO and GSDMD^KO mice. Interestingly, the addition of IFN-γ prolonged the survival of IRF2^KO mice during *F. novicida* infection, and even allowed ~75% of GSDMD^KO mice to survive the infection, indicating its therapeutic potential. This therapeutic potential is consistent with previous studies, including an in vivo intravenous model of infection with *Francisella* live vaccine strain (LVS) (Anthony et al, 1989) and a study in ASC^KO mice demonstrating that rIFN-γ can act independently of the inflammasome (Wallet et al, 2017).

The higher efficacy of rIFN-γ therapy in GSDMD^KO compared to IRF2^KO mice underscores the broader immune defect of this latter mouse line. Indeed, beyond NK cell deficiency, IRF2^KO mice exhibit defective Th1 cell differentiation, a feature critical for immune responses to *F. tularensis* (Duckett et al, 2005). Importantly, rIFN-γ therapy increased NK cell number in both IRF2^KO and WT mice. Although these NK cells displayed a less mature phenotype, they exhibited enhanced cytotoxic potential. Moreover, rIFN-γ treatment upregulated GSDMD expression, indicating that this therapy may be beneficial not only for patients suffering from primary immunodeficiencies but also for those without any genetic defects. Of note, primary immunodeficiencies resulting from loss of function in IRF2 have not yet been identified in humans. IRF2 expression is frequently lost in cancer cells (Kriegsman et al, 2019), and genetic variants in IRF2 impact immune responses (Fairfax et al, 2014), with variant associations reported for atopic dermatitis and eczema herpeticum (Gao et al, 2012). Importantly, rIFN-γ therapy (ACTIMMUNE® or Imukin®) is currently used for patients presenting chronic granulomatous disease (Mouy et al, 1991) and sepsis-induced immunosuppression (Payen et al, 2019). Our findings suggest that rIFN-γ therapy may also be beneficial (potentially as an adjuvant to antibiotics) in patients with severe infections caused by *F. tularensis* and potentially other intracellular bacterial pathogens.

Overall, the study highlights the multifaceted role of IRF2 in immune regulation, notably in inflammasome activation and NK cell maturation and acquisition of antibacterial effectors in the context of bacterial infection. The findings suggest potential therapeutic avenues for modulating these pathways, and especially the therapeutic potential of IFN-γ in the context of infectious diseases.

# Methods

**Reagents and tools table**

| Reagent/resource | Reference or source | Identifier or catalog number |
|---|---|---|
| **Experimental models** | | |
| Mouse: C57BL/6J | Charles River | N/A |
| Mouse: IRF2^KO | PBES (CRISPR/Cas9) | This study |
| Mouse: GSDMD^KO | Petr Broz, UNIL | Heilig et al, 2018 |
| Mouse: ASC^KO | Vishva Dixit, Genentech | Mariathasan et al, 2005 |
| Mouse: AIM2^KO | Vishva Dixit, Genentech | Jones et al, 2010 |
| Mouse: Caspase-11^KO | Junying Yuan | Wang et al, 1998 |
| Mouse: B6.SJL-PtprcaPepcb/BoyCrl (Ly5a, CD45.1) | Charles River | N/A |
| *Francisella novicida* strain U112 | Laboratory stock | N/A |
| *Francisella novicida* ΔFPI | Laboratory stock | N/A |
| *E. coli* J53 | Laboratory stock | N/A |
| **Recombinant DNA** | | |
| **Antibodies** | | |
| **Western blot** | | |
| Anti-GSDMD | Abcam | Ab209845 |
| Anti-Caspase-1 | Adipogen | AG-20B-0042-C100 |
| Anti-ASC | Cell Signaling | 67824 |
| Anti-β-actin | Sigma-Aldrich | MAB1501 |
| Anti-rabbit HRP | Sigma-Aldrich | A0545 |
| Anti-mouse HRP | Promega | W402.B |
| **Flow cytometry-surface and intracellular** | | |
| Live/dead | BD | 564996 |
| CD45 | BD | 563891 |
| CD122 | BD | 751560 |
| CD49b | BD | 741097 |
| KLRG1 | BD | 740279 |

| Reagent/resource | Reference or source | Identifier or catalog number |
|---|---|---|
| CD11b | BD | 566417 |
| CD27 | BD | 741518 |
| CD62L | BD | 553151 |
| CD49a | BD | 564863 |
| CD69 | BD | 747481 |
| CD19 | BD | 553785 |
| Tcf7 | BD | 566784 |
| TCRγδ | BD | 564874 |
| Ki67 | BD | 568148 |
| NK1.1 | BD | 564143 |
| Ly49H | BioLegend | 144724 |
| NKp46 | BioLegend | 137637 |
| CD127 | BioLegend | 135016 |
| CD3 | BioLegend | 100306 |
| Ly6G | BioLegend | 108404 |
| Strepta | BioLegend | 405227 |
| NKG2D | ebiosciences | 61-5882-82 |
| CD25 | ebiosciences | 35-0251-82 |
| IL18Ra | Thermo | 48-5183-80 |
| F4/80 | Thermo | 11-4801-85 |
| Eomes | Thermo | 46-4875 |
| Tbet | Thermo | 606-5825-82 |
| GzmA | Thermo | 25-5831-82 |
| **Oligonucleotides and other sequence-based reagents** | | |
| IRF2 genotyping primer F | GCATCTGGGCCTACAGCTAG | This study |
| IRF2 genotyping primer R | CTGGGCCCACTTTCATCAAG | This study |
| Gsdmd qPCR F | CCGGGTTGAGCAGACAATAG | This study |
| Gsdmd qPCR R | ACCACTTTCTCAAAGGCCG | This study |
| β-actin qPCR F | CTAAGGCCAACCGTGAAAAG | This study |
| β-actin qPCR R | ACCAGAGGCATACAGGGACA | This study |
| crRNA1 | AACGGAUGCGAAUGCGCCCG | IDT |
| crRNA2 | AUAAAUUCCAAUACGAUACC | IDT |
| **Chemicals, enzymes and other reagents** | | |
| LPS (E. coli O111:B4, Ultrapure) | InvivoGen | tlrl-eblps |
| Gentamicin | Thermo Fisher | 15750-037 |
| DMEM-GlutaMAX | Thermo Fisher | 61965-026 |
| Lympholyte | Cedarlane | CL5031 |
| FBS | Dutscher | 500105 |
| M-CSF (L929 supernatant) | Laboratory source | N/A |
| RIPA buffer | Various | N/A |
| Foxp3 Fix/Perm kit | eBioscience | 00-5523-00 |
| ACK lysing buffer | Gibco | A1049201 |
| Percoll | Cytiva | 17089102 |

| Reagent/resource | Reference or source | Identifier or catalog number |
|---|---|---|
| LDH Cytotoxicity Kit | Promega | CytoTox 97 |
| IL-15/IL-15R complexes | eBiosciences | 14-8152-80 |
| Recombinant IL-18 | Cliniciences | B004-5 |
| Recombinant IFN-γ | ImmunoTools | 12343537 |
| IL-2 (muIL2 supernatant) | Laboratory source | N/A |
| **Software** | | |
| FlowJo | Tree Star | N/A |
| GraphPad Prism | GraphPad | N/A |
| RStudio | RStudio | N/A |
| OMIQ | Dotmatics | N/A |
| Compass Simple Western | ProteinSimple | v6.1.0 |
| CRISPOR | Online tool | N/A |
| **Other** | | |
| Aurora Flow Cytometer | Cytek | N/A |
| LSRII/Fortessa | BD Biosciences | N/A |
| Jess Simple Western System | ProteinSimple | N/A |
| Bio-Plex MAGPIX | Bio-Rad | N/A |
| NEPA 21 Electroporator | Nepa Gene | N/A |
| Precellys Evolution | Bertin Technologies | N/A |
| CytoTox96 LDH kit | Promega | G1780 |
| Mouse IL-18 Platinum ELISA kit | eBiosciences | BMS618 |
| Mouse IFN-γ DuoSet ELISA | R&D | DY485 |
| Mouse IL-1β DuoSet ELISA | R&D | DY401 |
| Mouse TNF DuoSet ELISA | R&D | DY410 |
| ProcartaPlex Cytokine Panel | Thermo Fisher | PPX-15 |
| ImProm-II Reverse Transcriptase | Promega | A3802 |
| FastStart Universal SYBR Green Master (Rox) | Roche | 3736468 |

## Ethics statement

This study adhered to the French recommendations outlined in the Guide for the Ethical Evaluation of Experiments Using Laboratory Animals (http://gircor.net/qui/ethicalEvaluationGuide4LaboratoryAnimals.pdf) and the European guidelines 86/609/CEE. All animal experiments were reviewed and approved by the animal ethics committee (CECCAPP, Lyon, Ministère de l'Enseignement Supérieur de la Recherche #C2EA15), France, under protocol number #ENS_2022_012.

## Mice, animal infection, and LPS injection

All animals were housed in a dedicated animal facility (Plateau de Biologie Expérimentale de la Souris, Lyon, France). *Irf2^KO* mice were generated and bred by the animal facility PBES, using the

CRISPR-Cas9 technique, as previously described (Teixeira et al, 2018). Briefly, *IRF2* exon 2 was targeted using two sgRNA (5'-AACGGATGCGAATGCGCCCG-3' and 5'-TAAATTCCAATAC-GATACCA-3') designed using CRISPOR software 2. The corresponding crRNA (Eurogentec) were combined with TracrRNA using equimolar concentration and recombinant Cas9 protein (PNA Bio, #CP02) and electroporated into C57BL/6 J mouse pronuclear-stage embryos with intact zona pellucida using NEPA 21 electroporator. The embryos that reached the two-cell stage were transferred into the oviduct of B6CBAF1 (Charles River, France) pseudopregnant females. *IRF2^KO* animals were identified by PCR using the following primers 5'-GCATCTGGGCCTACAGCTAG-3'; 5'-CTGGGCCCACTTTCATCAAG-3'. A strain with a 98-nucleotide deletion was selected (Fig. EV1). Sex- and age-matched littermates (IRF2 WT) were used as controls. The Ly5a mice (B6.SJL-*PtprcaPepcb*/BoyCrl) used in this study were purchased from Charles River Laboratories. ASC^KO and AIM2^KO mice were a gift from Vishva DIXIT at Genentech Inc. GSDMD^KO mice were a gift from Petr BROZ at UNIL and have been previously described (Heilig et al, 2018). Caspase-11^KO mice were a gift from Junying YUAN (Harvard Medical School, Boston) (Wang et al, 1998).

Mice (IRF2^WT, IRF2^KO, and GSDMD^KO), either male or female, aged 8–12 weeks, were infected subcutaneously with either 1000 or 5000 CFU of wild-type *Francisella novicida* (*F. novicida*) strain U112 in 100 µL PBS. Mice were euthanized when reaching predefined endpoints or at the desired time point.

In separate experiments, mice of the indicated genotypes were injected intraperitoneally with lipopolysaccharide (LPS, 3 µg per gram of body weight) (LPS-EB ultrapure (LPS from *E. coli* O111:B4), tlrl-eblps, Invivogen) and sacrificed 6 h post injection to collect blood for serum IL-18 quantification.

## Cell culture and infection

Bone marrow-derived macrophages (BMDMs) were derived from progenitors collected from mouse femurs, cultured for 6 days in DMEM-GlutaMAX (61965-026, Thermo Fisher) supplemented with 10% v/v FBS, 10% v/v M-CSF-containing L929 supernatant, pyruvate 1 mM (11360039, Thermo Fisher), Hepes 10 mM (15630-056, Thermo Fisher) and then collected for immediate use. Cells were seeded at $1.10^5$ cells per well in 96-well plates and kept at 37 °C, 5% $CO_2$. BMDMs were then infected with *F. novicida* U112 (WT or ΔFPI) or *E. coli* J53 (WT) at the indicated multiplicity of infection. BMDMs were then centrifuged at 1000×*g* for 20 min at 20 °C. Gentamicin (15750-037, Thermo Fisher) was added at 5 µg/mL 1-h post invasion to eliminate extracellular bacteria. Supernatants were collected at 6, 8, 24-h (*F. novicida*), or 20-h (*E. coli*) post invasion.

## Coculture of BMDMs with splenic mononuclear cells

Two coculture systems were established to segregate the genetic contribution of infected cells (mostly phagocytes) from the genetic contribution of IFN-γ-producing cells (mostly NK cells). A 70-µm cell strainer was used to generate a single-cell suspension of splenic cells. Splenic mononuclear cells were then isolated by density gradient centrifugation of the spleen cell suspension using lympholyte cell separation media (Cedarlane, CL5031). Lympholyte

was gently added using a Pasteur pipette, allowing it to layer beneath the cell suspension by gravity. The sample was then centrifuged at 1800 rpm for 20 min at 20 °C. After centrifugation, the cells at the interface were carefully collected and transferred to a new tube for washing.

a) Coculture of WT, IRF2^KO, or GSDMD^KO BMDMs with WT splenic mononuclear cells: WT, IRF2^KO, or GSDMD^KO BMDMs were seeded at $1 \times 10^5$ cells per well in a 96-well plate and infected with *F. novicida* U112 (WT, MOI1, and MOI10). Gentamicin (15750-037, Thermo Fisher) at 5 µg/mL was added 1-h post-invasion, along with WT splenic mononuclear cells, treated or untreated with IL-15/IL-15R complexes (10 ng/mL) (Peprotech), IL-18 (200 ng/mL) (Cliniciences), or both IL-15/IL-15R complexes and IL-18 (Marcais et al, 2014).

b) Coculture of WT BMDMs with WT, IRF2^KO, or GSDMD^KO splenic mononuclear cells: WT BMDMs were seeded at $1 \times 10^5$ cells per well in a 96-well plate and infected with *F. novicida* U112 (WT, MOI1, and MOI10). Gentamicin (15750-037, Thermo Fisher) at 5 µg/mL was added 1-h post-invasion, along with WT or IRF2^KO or GSDMD^KO splenic mononuclear cells, treated or untreated with IL-15/IL-15R complexes, IL-18, or both IL-15/IL-15R complexes and IL-18.

Supernatants were collected 24 h later, and the level of IFN-γ cytokine was measured via ELISA.

## Purification and differentiation of NK cells into LAK cells for in vivo injection

NK cells were isolated by negative selection using a cocktail of biotinylated antibodies and anti-biotin microbeads (Miltenyi), and magnetic separation using an AutoMACS (Miltenyi), resulting in a purity ranging from 60 and 80%. The purified NK cells were cultured in medium supplemented with 10% supernatant of IL-2-producing cells (muIL2, (Karasuyama and Melchers, 1988)). After 7 days of culture, these lymphokine-activated killer (LAK) cells (>98% NK cells) were used for in vivo transfer. On day 0, IRF2^KO mice were intravenously injected with a suspension of LAK cells ($10 \times 10^6$ cells in 100 µL HBSS) into the retro-orbital sinus. Control mice received 100 µL of HBSS alone. Eight hours later, the mice were infected with *F. novicida* as described previously.

## Automated and conventional Western blotting

### Automated Western immunoblotting

The Jess^TM Simple Western system (ProteinSimple, San Jose, CA, USA) is a size-based automated capillary Western blot assay. To quantify inflammasome components, we followed the manufacturer's standard method for the 12–230-kDa Jess separation module (SM-W004). BMDM from different genotypes were lysed using RIPA buffer (1% NP40 v/v, 0.5% sodium deoxycholate w/v and 0.1% sodium dodecyl sulfate w/v in PBS, pH 7.4). The lysates were mixed with 0.1X Sample buffer (ProteinSimple) to reach a final concentration of 1 µg/µL and then were mixed with the Fluorescent 5X Master mix (ProteinSimple), containing 200 mM dithiothreitol (ProteinSimple). The lysates were denatured at 95 °C for 5 min and then loaded onto the plate in the presence of a fluorescent

molecular weight marker (12–230-kDa PS-ST01EZ). Proteins were separated in capillaries under 375 volts migration and incubated with the relevant primary antibodies (see below). After a wash step, either goat anti-rabbit secondary HRP antibody (DM-001) or goat anti-mouse secondary HRP antibody (DM-002) (ProteinSimple) was used, followed by incubation with streptavidin-HRP and revelation using Luminol-Peroxide Mix (ProteinSimple). Digital images of the chemiluminescence in the capillary were captured using Compass Simple Western software (version 6.1.0, Protein Simple), which automatically calculated heights (chemiluminescence intensity), area, and signal-to-noise ratio. Rabbit-anti-GSDMD (ab209845, Abcam), mouse-anti-Caspase-1 (C1 p20 Casper1, AG-20B-0042-C100, Adipogen) and rabbit-anti-ASC (ASC/TMS1 (D2W8U), 67824, Cell Signaling) antibodies were used.

### Conventional Western blotting

Western blot analysis was performed on in vivo tissue samples to detect GSDMD, Caspase-1, and β-actin proteins. To obtain lysates, one volume of samples was treated with one volume of RIPA buffer supplemented with complete, EDTA-free protease inhibitor cocktail (Roche). Suspensions were clarified by centrifugation at 13,000 rpm for 5 min, at 4 °C. Samples were incubated with Laemmli Sample buffer supplemented with β-Mercaptoethanol for 10 min at 95 °C before performing SDS polyacrylamide gel electrophoresis (PAGE).

Proteins were transferred to polyvinylidene difluoride (PVDF) membranes. The membranes were blocked with 4% BSA and 0.1% Tween®20 in Tris Buffered Saline (TBS). GSDMD, Caspase-1 and β-actin proteins were immunolabeled. To this end, membranes were incubated with rabbit-anti-GSDMD (ab209845, Abcam), mouse-anti-Caspase-1 (C1 p20 Casper1, AG-20B-0042-C100, Adipogen), or mouse-anti-β-actin (clone C4, MAB1501, Sigma-Aldrich) antibodies diluted in TBS. After washing with wash buffer (0.1% Tween®20 in TBS), membranes were incubated with anti-rabbit-HRP (A0545, Sigma-Aldrich) or anti-mouse-HRP (W402.B, Promega) secondary antibodies diluted in TBS. Finally, membranes were washed as before, treated for 1 min with GE Healthcare ECL™ (Merck) and revealed.

### Immune cell phenotypic characterization and OMIQ analysis

Organs were collected aseptically and processed to isolate single-cell suspensions. Spleen and liver were mechanically disrupted and filtered through a sterile 100-μm nylon mesh filter (BD Biosciences). Lungs were enzymatically dissociated with the Lung dissociation kit according to the manufacturer's instructions (Miltenyi Biotec). Blood samples (100 μl) were collected on EDTA by retro-orbital bleeding and treated with BD Facs Lysing solution (BD Biosciences) after flow-cytometry staining. Bone marrows were flushed and vigorously resuspended to generate single-cell suspensions. Single cell suspensions from organs were isolated and enriched for mononuclear cells using Percoll gradient (liver and lung) or not (spleen and bone marrow). Red blood cells were lysed using ACK Lysing Buffer (Gibco) during 2 min on ice. Cell numbers were calculated using calibrated fluorescent beads (flow-count fluorospheres, Beckman Coulter) or using the Accuri C6 Cytometer (BD Biosciences).

Cells were stained with appropriate antibodies for flow cytometry analysis to characterize different cell populations. Briefly, surface staining was performed on single-cell suspensions for 30 min at 4 °C with the appropriate mixture of Abs diluted in staining buffer (PBS, 1% FCS [Life Technologies] and 0.09% NaN₃ [Sigma-Aldrich, Saint Quentin-Fallavier, France]). The following Abs (dilution, clones) were used: anti-CD45 (200, 30-F11), CD3ε (300, 2c11), CD4 (200, RM4.5), CD8 (200, 53-6.7), CD11b (400, M1/70), CD25 (200, PC61), CD45R/B220 (300, RA3-6B2) (all BD PharMingen) and Ly6G (400, 1A8), Ly6C (200, AL-21), NK1.1 (200, PK136) (all Biolegend). Flow cytometry analyses were performed on a Becton Dickinson FACS Fortessa LSRII and analysed with FlowJo software (TreeStar, Ashland, OR, USA).

To characterize innate lymphocytes and their maturation, the following staining was performed: The live-dead fixable ViaDyeRed (1000e, Cytek®) was co-incubated with anti-TCRγδ (100e, GL3). Samples were stained for 30 min at 4 °C and then washed with PBS. Membrane staining was performed in PBS for 30 min at 4 °C with anti-CD45 (100, 30-F11), anti-CD122 (100, 5H4), anti-NK1.1 (100, PK136), anti-NKp46 (100, 29A1.4), anti-CD49b (200, DX5), anti-KLRG1 (100, 2F1), anti-CD11b (200, M1/70), anti-CD27 (200, LG.3A10), anti-CD127 (50, A7R34), anti-CD49a (100, Ha31/8), anti-IL18Ra (100, P3TUNYA), anti-CD69 (100, H1.2F3), anti-CD19 (200, 1D3), anti-CD3 (200, 145-2C11), anti-F4/80 (200, BM8), anti-Ly6G (200, RB6-8C5). Samples were washed with PBS and then fixed with the Foxp3 fixative buffer kit for 1 h. Samples were washed with Foxp3 PermWash Buffer (eBiosciences). Intracellular markers were then stained for 1 h at room temperature with Streptavidin (100, BLE 405227), anti-Eomes (400, Dan11mag), anti-Tbet (200, 4B10), anti-Granzyme A (400, Gza-3G8.5), and anti-Ki67 (400, B56). Samples were analyzed on an Aurora flow cytometer (Cytek®). Antibodies were purchased from eBioscience, BD Biosciences or Biolegend. Data were analysed using FlowJo (Treestar®), RStudio and OMIQ (Dotmatics®), a flow cytometry online analysis platform used to run higher-level visualizations such as t-distributed stochastic neighbor embedding (t-SNE).

### LDH assay

LDH activity in the cell supernatant was assessed as an indicator of cell death, using the CytoTox96 LDH kit (Promega, Charbonnières-les Bains) following the manufacturer's instructions. Once developed, the plate was read at 490 nm (Tecan). To normalize for spontaneous lysis, the percentage of LDH release was calculated using the formula: ((LDH infected sample – LDH untreated sample)/(LDH Triton-treated sample – LDH untreated sample)) *100.

### ELISA and Luminex

#### ELISA

mIL-1β, mTNF, mIL-18, and mIFN-γ cytokine release were measured using enzyme-linked immunosorbent assays in BMDM supernatant (mIL-1β, mIL-18 and mTNF), coculture of BMDMs with splenic mononuclear cells (mIFN-γ) or mice sera (mIL-18 and mIFN-γ), respectively. Blood was collected at either 16-h post invasion or 48-h post invasion and centrifuged to obtain serum. The following kits were used: Mouse IL-18 Platinum ELISA kit (BMS618, eBiosciences), Mouse IFN-γ DuoSet ELISA (DY485,

R&D), Mouse IL-1β DuoSet ELISA (DY401, R&D), and Mouse TNF DuoSet ELISA (DY410, R&D).

### Luminex

The concentrations of IFN alpha, IFN beta, IFN gamma, IL-1 alpha, IL-1 beta, IL-4, IL-5, IL-6, IL-10, IL-12p70, IL-13, IL-15/IL-15R, IL-17 (CTLA-8), IL-18, and TNF alpha in mouse sera were assessed using ProcartaPlex assay (Thermo Fisher). Assay was run following the manufacturer's guidelines for mouse sera. Data were acquired on the Bio-Plex MAGPIX Multiplex Reader.

## Bacterial proliferation assay

Lungs, spleen, and liver were collected 48-h post infection. Organs were grinded using a Precellys evolution tissue homogenizer (Bertin Technologies) in PBS. Crushed organs were then spread on TSA + Cysteine 0.1% + Ampicillin 100 µg/mL using easySpiral Dilute (Intersciences) automated diluter and plater and stored at 37 °C, 5% $CO_2$ for 48 h. Bacterial colony counting was performed using the Scan-1200 automated colony counter and its associated software (Intersciences).

## In vivo effects of rIFN-γ in mice

Mice were injected with 10 µg of rIFN-γ (Immunotools) subcutaneously at day 0 along with *F. novicida* (Wallet et al, 2017). For the next 4 days (days 1–4), mice received a daily intraperitoneal injection of rIFN-γ (10 µg in 100 µL PBS). The experiment was terminated either upon reaching predefined endpoints or at the desired time point for analysis.

## Bone marrow chimera experiments

Competitive bone marrow transplantation experiments were conducted using the following methodology. Irradiated (9 Gy using an X-ray irradiator X-RAD 320) Ly5a homozygous mice with the CD45.1 allele were used as recipients. Two types of donor cells were used: Ly5a heterozygous (Het) mice with both CD45.1 and CD45.2 alleles (designated as CD45.1/CD45.2 or WT) and IRF2 homozygous mice with the CD45.2 allele. Recipient mice were divided into two groups: (i) reconstituted with a mixture of 50% Ly5a Het and 50% IRF2$^{KO}$ donor cells (50% WT/50% IRF2$^{KO}$), or (ii) reconstituted with 100% Ly5a Het donor cells only (100% WT). Donor cell reconstitution was quantified using allele-specific CD45 antibodies, which allowed the distinction between CD45.1- and CD45.2- expressing cells.

## RNA and quantitative real-time RT-PCR

Total RNA was extracted. To this end, trizol was added to 500 µL organ lysate to obtain a final volume of 1 mL. Samples were incubated for 5 min at 65 °C, 200 µl chloroform was added, and samples were centrifuged (13,000 rpm, 15 min, 4 °C). The aqueous phase was collected and washed once with 120 µl chloroform. The aqueous phase was further incubated with 500 µl isopropanol for 10 min at room temperature. Then, the samples were centrifuged at 13,000 rpm for 10 min (4 °C), and the supernatant was discarded. After washing the pellet twice with 70% ethanol, the pellets were dried, and RNA was resuspended in the appropriate volume of RNase-free water.

RNA reverse transcription was performed using an Improm (Promega) kit. About 1 µl random primer and 3 µl RNase-free water were added to 1 µl RNA. The mixture was incubated for 5 min at 70 °C and 5 min at 4 °C. Thereafter, 6.6 µl RNase-free water, 4 µl reagent buffer, 2.4 µl $MgCl_2$, 1 µl deoxyribose nucleoside triphosphates (dNTPs), and 1 µl reverse transcriptase were added to a final volume of 15 µl. Reverse transcription occurred on a thermocycler with the following program: 5 min at 25 °C, 1 h at 42 °C, 15 min at 70 °C. After reverse transcription, 200 µl RNase-free water was added to the resulting cDNA.

A LightCycler® 480 SYBR Green I Master (Roche) was used for qPCR. 5 µl cDNA was mixed with 4.6 µl water, 0.2 µl forward primer, 0.2 µl backward primer and 10 µl reaction mix. Adequate primers to amplify GsdmD and β-actin mRNA were chosen: for GsdmD 5'-CCGGGTTGAGCAGACAATAG-3' and 5'-ACCACTTTCTCAAAGGCCG-3', and for β-actin 5'-CTAAGGC-CAACCGTGAAAAG-3' and 5'-ACCAGAGGCATACAGGGACA-3'. qPCR was measured on a LightCycler® 480 Instrument (Roche), acquiring the cycle threshold ($C_T$) value as an output. To obtain the $\Delta C_T$ value of a certain mRNA-of-interest, the housekeeping gene β-actin $C_T$ value was subtracted from the mRNA-of-interest $C_T$ value. Subsequently, the expression fold change was assessed by calculating $2^{-\Delta CT}$.

## Statistical analysis

All analyses were performed using GraphPad Prism (GraphPad Software, La Jolla, CA, USA, www.graphpad.com). Data were expressed as mean ± SEM (standard error of the mean). The Shapiro–Wilk test was used to assess dataset normality. Significant differences were assessed by (i) one-way ANOVA followed by post hoc parametric Tukey test, Dunnett test or Sidak test for multiple comparisons if data followed a normal distribution, and (ii) Kruskal–Wallis test followed by post hoc non-parametric Dunn's tests if the values did not follow a normal distribution. To compare survival curves, the log-rank test was used. Whenever a statistical difference or *p* value is not shown, the comparison was either not statistically significant or not tested.

## Declaration of generative AI and AI-assisted technologies in the writing process

During the preparation of this work, the authors used ChatGPT to proofread the manuscript. After using this tool/service, the authors reviewed and edited the content as needed and take full responsibility for the content of the publication.

# Data availability

All data were available within the article.

The source data of this paper are collected in the following database record: biostudies:S-SCDT-10_1038-S44319-026-00698-4.

## Peer review information

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

## Acknowledgements

We acknowledge the contribution of SFR Biosciences (UMS3444/CNRS, US8/Inserm, ENS de Lyon, UCBL) facilities "Le Plateau de Biologie Expérimentale de la Souris" (Jean-François HENRY and Géraldine CLAVEAU) (AniRA-PBES), AniRA-Cytométrie and Protein Sciences Facility (PSF) for assistance. This project was funded by a grant from Agence Nationale de la Recherche (ANR) (ANR-22-ASTR-0036-02; ICY-TULA). We thank Vishva DIXIT at Genentech Inc. for providing the ASC[KO] and AIM2[KO] mice, Petr BROZ at UNIL for providing GSDMD[KO] mice, and Junying YUAN (Harvard Medical School, Boston) for providing Caspase-11[KO] mice. We thank Suzy MARKOSSIAN for developing the IRF2[KO] mouse model we used, using CRISPR/Cas9 genome editing technology.

## Author contributions

**Maxence Cornut**: Investigation. **Sophia Djebali**: Investigation. **Elena Rondeau**: Investigation. **Sarah Dayet**: Investigation. **Theo Fayolle**: Investigation. **Julie Haagen**: Investigation. **Lucie Fallone**: Investigation. **Noëmi Rousseaux**: Investigation. **Emmanuelle Caspar**: Investigation. **Melissa Marcotte**: Investigation. **Amandine Martin**: Investigation. **Elise Courteboeuf**: Investigation. **Maelan Deschamps-Biboulet**: Investigation. **Marie Teixeira**: Investigation. **Jacqueline MARVEL**: Investigation. **Benedicte, F Py**: Investigation. **Thierry Walzer**: Investigation. **Antoine Marcais**: Investigation. **Thomas Henry**: Investigation. **Emilie Bourdonnay**: Investigation.

Source data underlying figure panels in this paper may have individual authorship assigned. Where available, figure panel/source data authorship is listed in the following database record: biostudies:S-SCDT-10_1038-S44319-026-00698-4.

## Disclosure and competing interests statement

The authors declare no competing interests.

# Expanded View Figures

**Figure EV1.  IRF2$^{KO}$ mice exhibit high susceptibility to *Francisella novicida* infection.**

(A) Schematic representation of the CRISPR-Cas9–engineered 98-nucleotide deletion in the IRF2 locus. The deleted region, spanning exon 2 (green arrow) and part of intron 2–3, is highlighted in the blue text box. The two sgRNA target sites used for genome editing are indicated in purple. (B) Representative genotyping results showing WT, heterozygous, and homozygous IRF2$^{KO}$ mice. (C) Immunoblot analysis confirming the efficient deletion of IRF2 in bone marrow-derived macrophages (BMDMs) from IRF2$^{KO}$ mice compared to wild-type (WT) controls. (D) IRF1 protein expression in spleens from WT ($n = 6$) and IRF2$^{KO}$ ($n = 6$) mice, assessed by flow cytometry. (E, F) *IRF1* transcript levels in spleens (E; WT $n = 11$, IRF2$^{KO}$ $n = 6$) and BMDMs (F, $n = 4$ per genotype). *GSDMD* transcripts were also measured as a control. (G) *GSDMD* mRNA levels in the liver, lung and spleen from WT mice under naive (PBS) or infected with *Francisella* conditions. At least nine mice per group are shown. Each symbol represents the value of an individual mouse; the bars indicate the mean with SEM. Statistical analysis: (E) Kruskal–Wallis analysis with Dunn's correction, (G) one-way ANOVA analysis with Tukey correction. * indicates $p < 0.05$; exact $p$ values are listed below. $P$ values: (E) 0.0149; (G) 0.0141.

▶

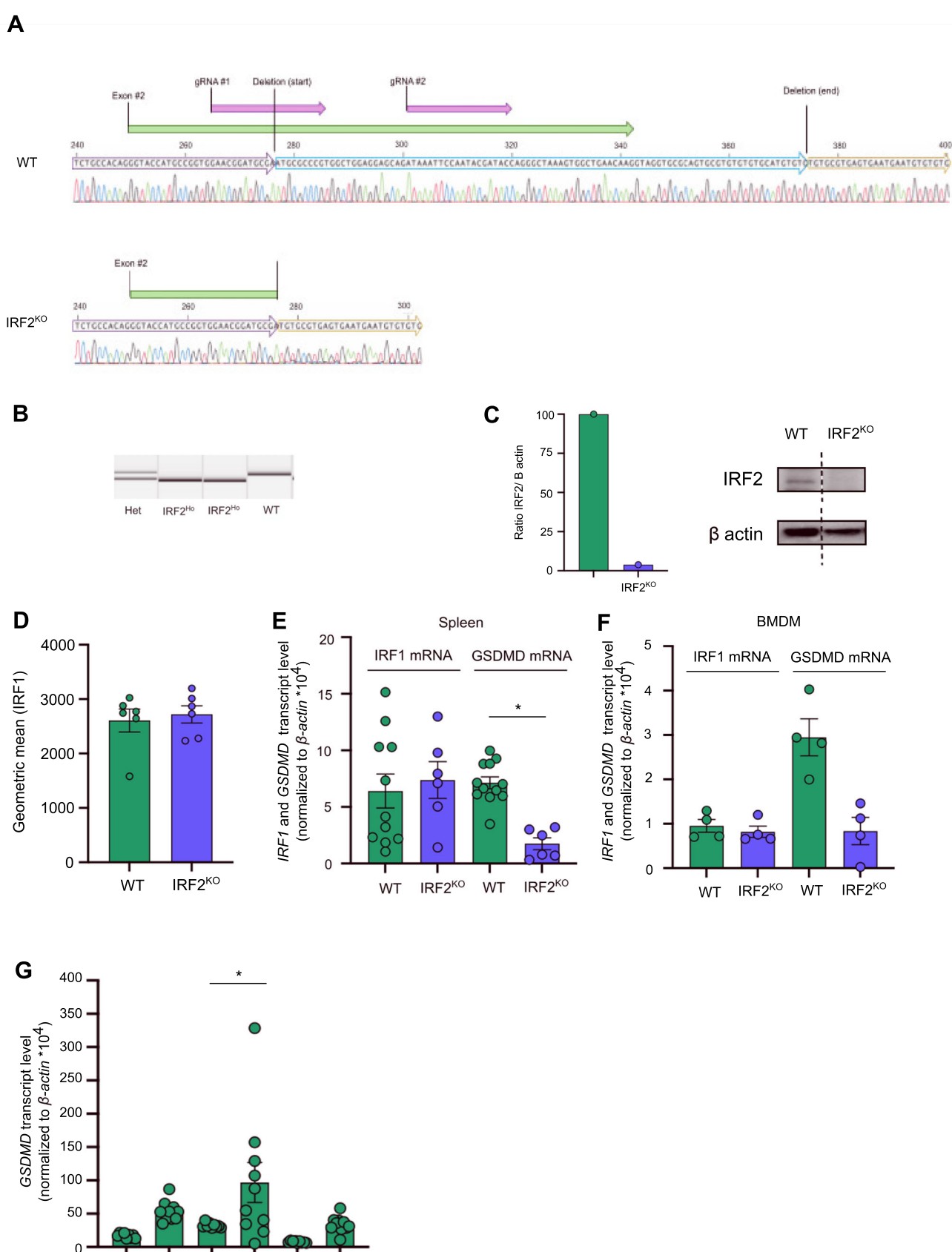

**A**

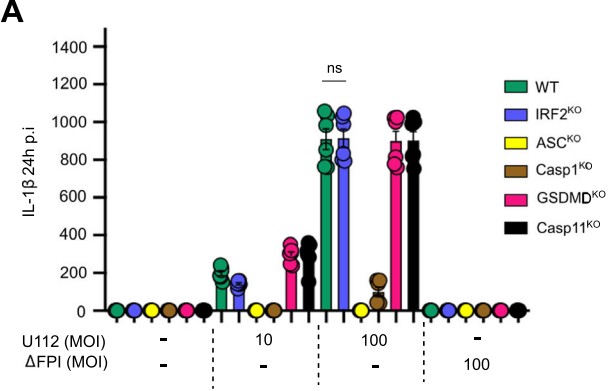

**B**

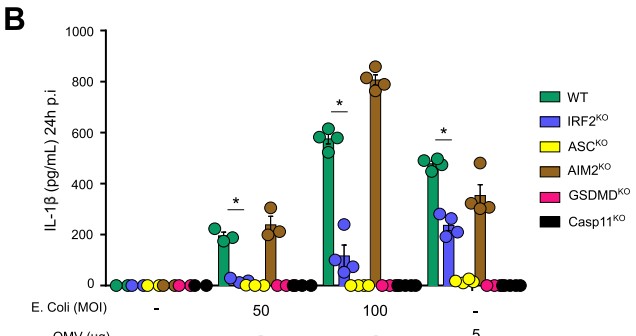

**C**

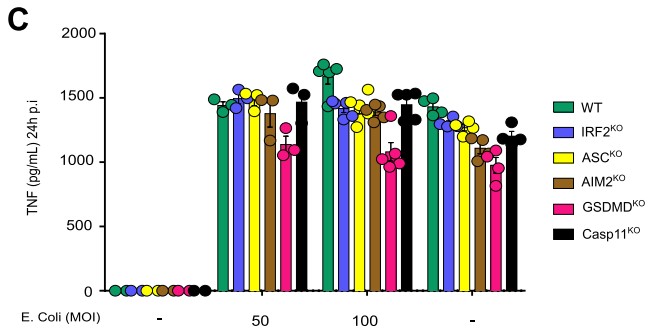

**D**

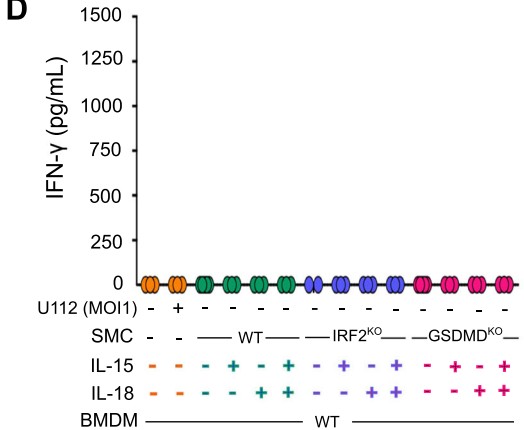

**E**

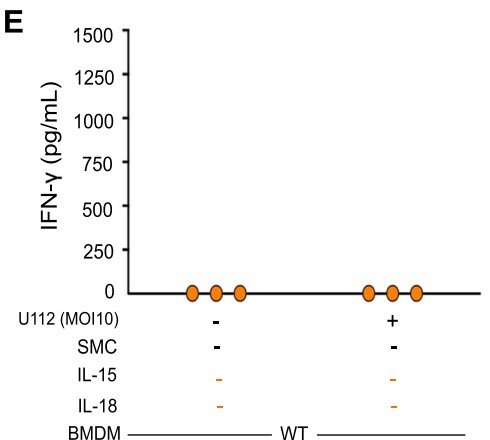

**F**

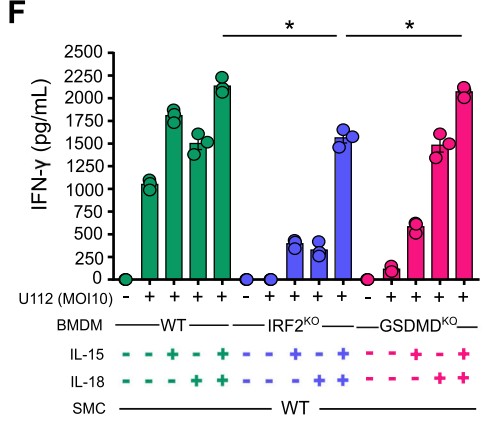

**G**

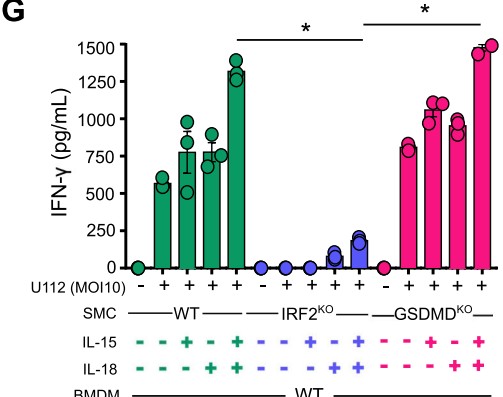

**Figure EV2. IRF2$^{KO}$ BMDMs are deficient in inflammasome activation in response to *F. novicida* and *E. coli* infection.**

IRF2 controls IFN-γ production through a dual regulation in infected cells and IFN-γ-producing cells. BMDMs from the indicated genotypes were infected or not with *F. novicida* (**A**) or with *E. coli* at an MOI of 50 or 100, or with 5 ug of OMV derived from *E. coli* (**B**, **C**). (**A**, **B**) IL-1β and (**C**) TNF concentrations were determined by ELISA. One experiment representative of at least three experiments is shown. The mean and the standard error of the mean (SEM) are shown. One-way ANOVA analysis with Tukey correction (**A**) was performed. (**D**, **E**) WT BMDMs were infected or not with *F. novicida* at a MOI of 1 (**D**) or MOI 10 (**E**). Those data serve as controls for the experiments in Fig. 3, where BMDMs were co-cultured with SMCs from the indicated genotype, either alone, with IL-15/IL-15R complexes (10 ng/mL), with IL-18 (200 ng/mL) or with both IL-15/IL-15R complexes and IL-18. Supernatants were collected 24 h later, and the level of IFN-γ cytokine was measured by ELISA. One experiment representative of at least three experiments is shown. (**F**) BMDMs from the indicated genotypes were infected or not with *F. novicida* at an MOI of 10, and then co-cultured with WT SMCs in the presence or not of IL-15/IL-15R complexes (10 ng/mL), IL-18 (200 ng/mL), or both IL-15/IL-15R complexes and IL-18. (**G**) WT BMDMs were infected or not with *F. novicida* at a MOI of 10, and then co-cultured with SMCs of the indicated genotypes in the presence or not of IL-15/IL-15R complexes (10 ng/mL), IL-18 (200 ng/mL), or both IL-15/IL-15R complexes and IL-18. Supernatants were collected 24 h later, and the level of IFN-γ cytokine was measured by ELISA. One dot represents the value from one replicate, the bar represents the mean and the standard error of the mean (SEM) of three replicates from one experiment, representative of three independent experiments. One-way ANOVA with Tukey's test (**B**) or Dunnett's test (**F**, **G**) was performed. * indicates *p* < 0.05; exact *p* values are listed below. *P* values: (**B**) from left to right: <0.0001, <0.0001, <0.0001; (**F**) from left to right: <0.0001, <0.0001; (**G**) from left to right: <0.0001, <0.0001.

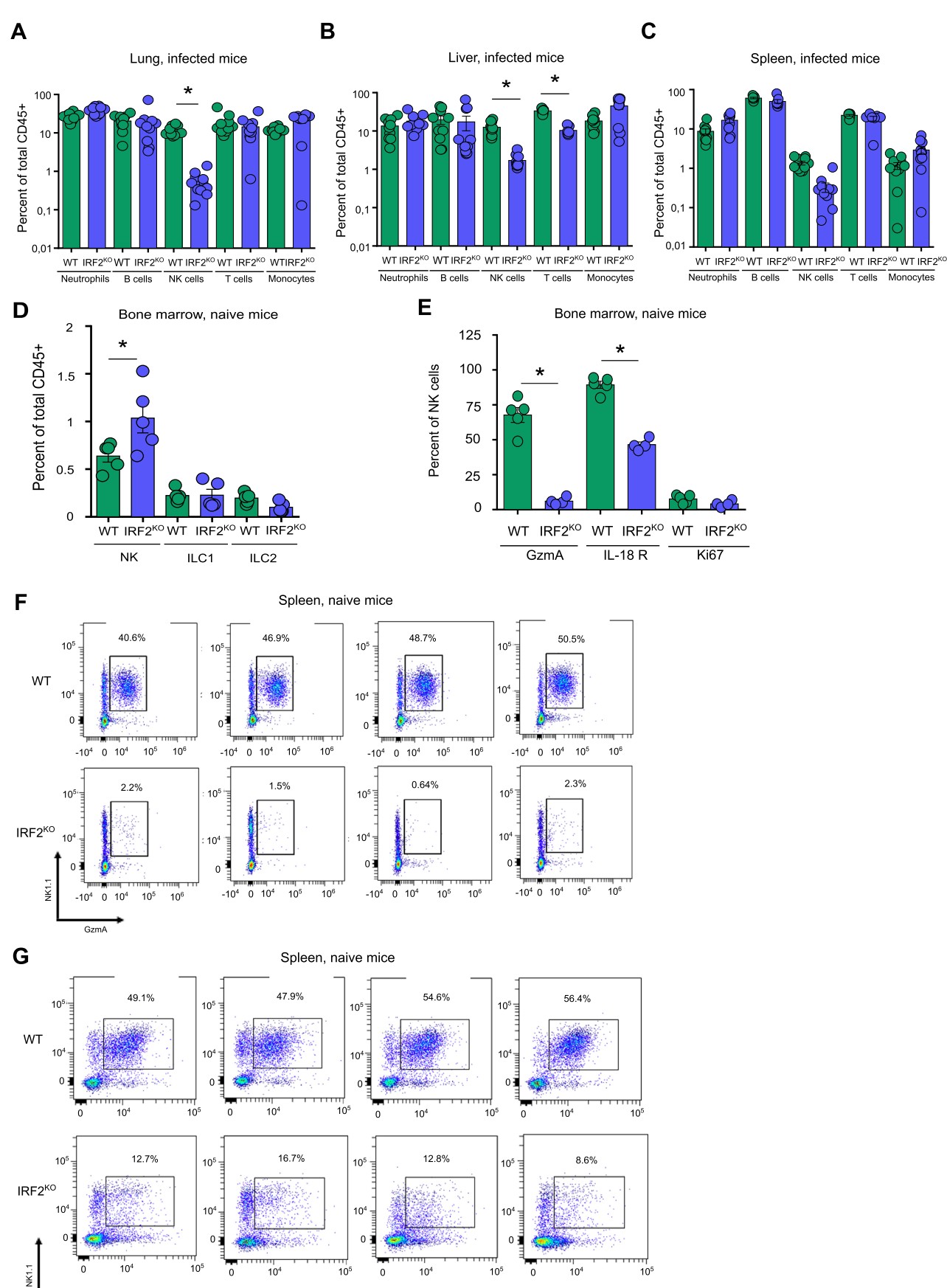

◀ **Figure EV3.   IRF2 regulates NK cell peripheral cell number and maturation, at steady state and during *F. novicida* infection.**

(A–C) WT or IRF2$^{KO}$ mice were infected with 2500 CFU for 2 days. (D–G) WT or IRF2$^{KO}$ naive mice are shown. (A–D) Frequency of CD45$^+$ subsets in the lung (A), the liver (B), the spleen (C) and in the bone marrow (D). (E) Frequency of GzmA$^+$, IL-18R$^+$, and Ki67$^+$ among total NK cells in the bone marrow from naive mice. (F, G) Flow cytometry dot plots comparing NK1.1 expression and GzmA (F) or IL18R (G) expression in spleen-derived NK cells from WT and IRF2$^{KO}$ mice. Each plot represents an individual sample, with the percentage of GzmA$^+$ or IL18R$^+$ NK cells indicated within the gated region. Each symbol represents the value of an individual mouse (A–E); the bars indicate the mean with SEM. The data were collected as follows: (A–C) show results from ten WT and ten IRF2$^{KO}$ mice; D shows results from 5 WT and 5 IRF2$^{KO}$ mice; (D) shows results from five WT and four IRF2$^{KO}$ mice; (F, G) show results from two WT and two IRF2$^{KO}$ mice. Kruskal–Wallis analysis with Dunn's correction was performed (A–C) and one-way ANOVA analysis with Tukey correction was performed (D, E). *P* values: (A) 0.0392, (B) from left to right: 0.0023, 0.0006; (D) 0.0115; (E) from left to right: <0.0001, <0.0001.

## A

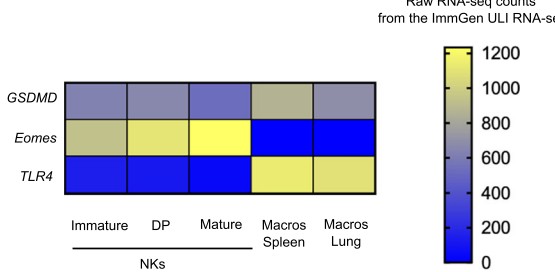

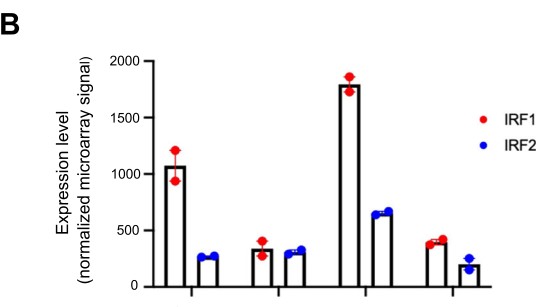

## B

**Figure EV4. GSDMD is expressed at low levels throughout the stages of NK cell differentiation.**

(A) The expression profile of *GSDMD*, in NK cells, splenic macrophages and alveolar macrophages, determined by RNAseq, is presented together with the expression profiles of *Eomes* and *TLR4*, as NK cell- and macrophage-specific positive controls, respectively (data extracted from ImmGen ULI RNAseq). (B) Expression levels of *IRF1* and *IRF2* in each specific organ is displayed according to BioGPS. Gene expression values were obtained from two independent publicly available BioGPS datasets. Bars indicate the mean with SEM.

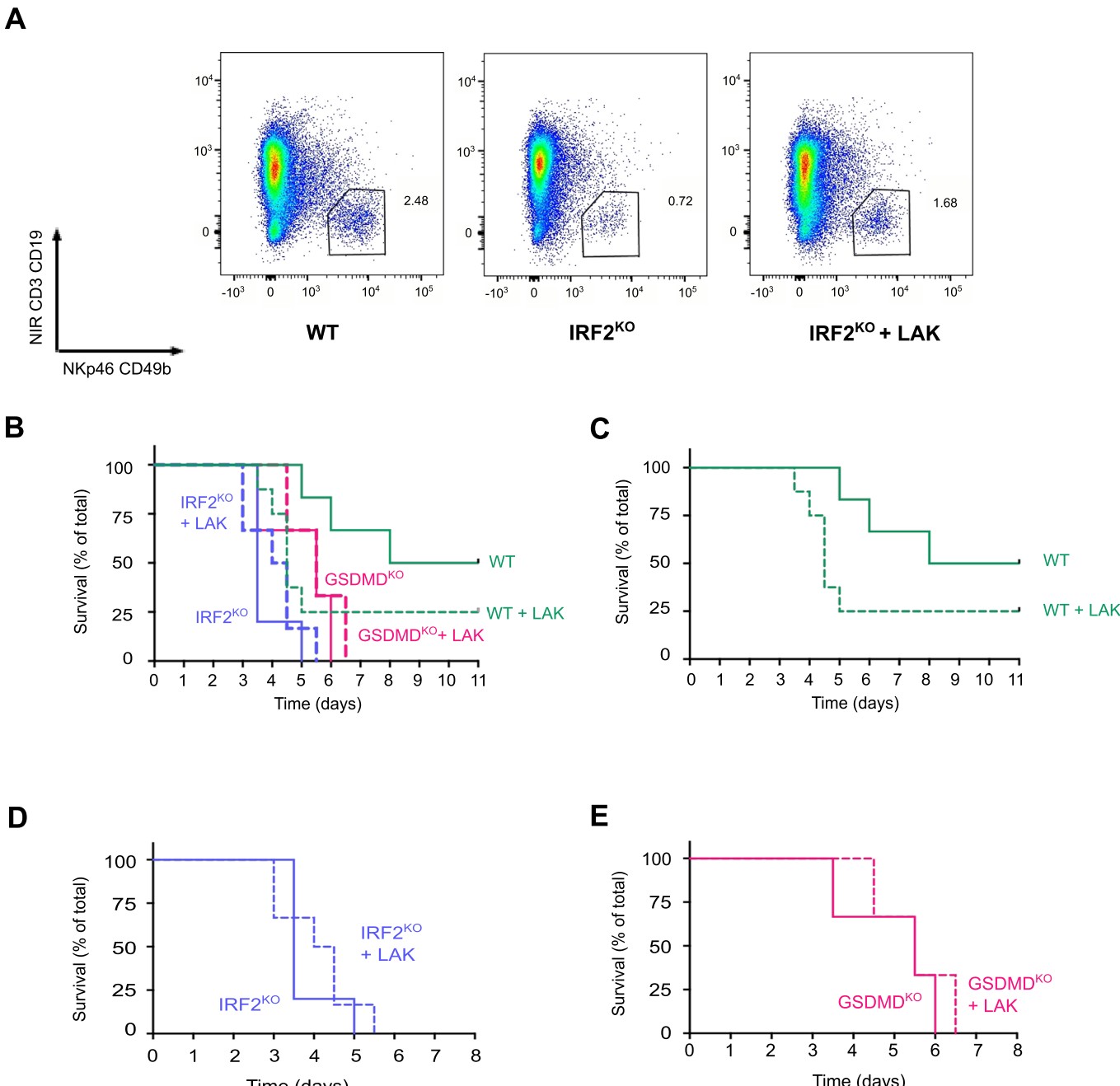

**Figure EV5. Survival of LAK-treated mice during *F. novicida* infection.**

(**A**) NK cells from Ly5a heterozygous (Het) mice with both CD45.1 and CD45.2 alleles (designated as CD45.1/CD45.2) were used to generate LAK cells, by incubating them with IL-2 for 7 days. Efficacy of the graft in IRF2$^{KO}$ recipient mice is shown. (**B–E**) Mice of the indicated genotypes received LAK cells or not, at day 0, and then were subcutaneously (sc) injected with $7 \times 10^2$ CFU at day 1. Survival was monitored twice a day. The data were collected as follows: (**B–E**) show results from six mice in the WT + PBS group, eight mice in the WT + LAK group, five mice in the IRF2$^{KO}$ + PBS group, six mice in the IRF2$^{KO}$ + LAK group, three mice in the GSDMD$^{KO}$ + PBS group, and three mice in the GSDMD$^{KO}$ + LAK group. Log-rank Cox–Mantel test (**B–E**) was performed.

