## [Peer Review File · EMBO Reports]

IRF2 deficiency disrupts pyroptosis, NK cell interferon- γ production and resistance to *Francisella*

Maxence Cornut, Sophia Djebali, Elena Rondeau, Sarah DAYET, Theo Fayolle, Julie Haagen, Lucie Fallone, Noëmi Rousseaux, Emmanuelle Caspar, Melissa Marcotte, Amandine Martin, Elise COURTEBOEUF, Maelan Deschamps-Biboulet, Marie Teixeira, Jacqueline MARVEL, Benedicte Py, Thierry Walzer, Antoine Marcais, Thomas Henry, and Emilie Bourdonnay

Corresponding author(s): Emilie Bourdonnay (emilie.bourdonnay@inserm.fr) , Thomas Henry (thomas.henry@inserm.fr)

Review Timeline:

Submission Date:	25th Feb 25
Editorial Decision:	12th May 25
Revision Received:	6th Nov 25
Editorial Decision:	8th Dec 25
Revision Received:	19th Dec 25
Accepted:	16th Jan 26

Editor: Achim Breiling

Transaction Report:

Dear Dr. Bourdonnay,

Thank you for the submission of your manuscript to EMBO reports. I have now received the reports from the three referees that were asked to evaluate your study, which can be found at the end of this email.

As you will see, the referees think that these findings are of interest. However, they have several comments, concerns, and suggestions, indicating that a major revision of the manuscript is necessary to allow publication of the study in EMBO reports. As the reports are below, and all the referee concerns need to be addressed, I will not detail them here. Please address the novelty concerns of referee #2 in the rebuttal letter.

Given the constructive referee comments, I would like to invite you to revise your manuscript with the understanding that the concerns of the referees must be addressed in the revised manuscript and/or in a detailed point-by-point response. Acceptance of your manuscript will depend on a positive outcome of a second round of review. It is EMBO reports policy to allow a single round of revision only and acceptance of the manuscript will therefore depend on the completeness of your responses included in the next, final version of the manuscript.

- 1) a .docx formatted version of the final manuscript text (including legends for main figures, EV figures and tables), but without the figures included. Figure legends should be compiled at the end of the manuscript text.
- 2) individual production quality figure files as .eps, .tif, .jpg (one file per figure), of main figures and EV figures. Please upload these as separate, individual files upon re-submission.

- 4) a complete author checklist, which you can download from our author guidelines

(<https://www.embopress.org/page/journal/14693178/authorguide>). Please insert page numbers in the checklist to indicate where the requested information can be found in the manuscript. The completed author checklist will also be part of the RPF.

5) that primary datasets produced in this study (e.g. RNA-seq, CHIP-seq, structural and array data) are deposited in an appropriate public database. If no primary datasets have been deposited, please also state this in a dedicated section (e.g. 'No primary datasets have been generated and deposited'), see below.

The accession numbers and database should be listed in a formal "Data Availability" section that follows the model below. This is now mandatory (like the COI statement). Please note that the Data Availability Section is restricted to new primary data that are part of this study. This section is mandatory. As indicated above, if no primary datasets have been deposited, please state this in this section

Data availability

6) We now request the publication of original source data with the aim of making primary data more accessible and transparent to the reader. You will receive a separate email with instructions for providing source data with your revised manuscript, including information how to upload and organize the files.

8) Regarding data quantification and statistics, please make sure that the number "n" for how many independent experiments were performed, their nature (biological versus technical replicates), the bars and error bars (e.g. SEM, SD) and the test used to calculate p-values is indicated in the respective figure legends (also for EV and Appendix figures). Please also check that all the p-values are explained in the legend, and that these fit to those shown in the figure. Please provide statistical testing where applicable. Please avoid the phrase 'independent experiment', but clearly state if these were biological or technical replicates. Please also indicate (e.g. with n.s.) if testing was performed, but the differences are not significant. In case n=2, please show the data as separate datapoints without error bars and statistics. See also: <http://www.embopress.org/page/journal/14693178/authorguide#statisticalanalysis>

9) Please add scale bars of similar style and thickness to microscopic images, using clearly visible black or white bars (depending on the background). Please place these in the lower right corner of the images themselves. Please do not write on or near the bars in the image but define the size in the respective figure legend.

10) Please also note our reference format:

12) We now use CRediT to specify the contributions of each author in the journal submission system. CRediT replaces the author contribution section. Please use the free text box to provide more detailed descriptions and do NOT provide your final manuscript text file with an author contributions section. See also our guide to authors: <https://www.embopress.org/page/journal/14693178/authorguide#authorshipguidelines>

13) All Materials and Methods need to be described in the main text using our 'Structured Methods' format, which is required for all research articles. According to this format, the Methods section should include a Reagents and Tools Table (listing key reagents, experimental models, software, and relevant equipment and including their sources and relevant identifiers), uploaded as separate file, and a Methods section in which we encourage the authors to describe their methods using a step-by-step protocol format with bullet points, to facilitate the adoption of the methodologies across labs. More information on how to adhere to this format as well as downloadable templates (.doc) for the Reagents and Tools Table can be found in our author guidelines (section 'Structured Methods'):

14) Please add up to five keywords to the manuscript and order the sections like this, using these names:
Title page - Abstract - Keywords - Introduction - Results - Discussion - Methods - Data availability section - Acknowledgements - Disclosure and Competing Interests Statement - References - Figure legends - Expanded View Figure legends

15) Please make sure that all the funding information is also entered into the online submission system and that it is complete and similar to the one in the acknowledgement section of the manuscript text file.

I look forward to seeing a revised form of your manuscript when it is ready.

Yours sincerely,

Referee #1:

In this manuscript "IRF2 deficiency weakens resistance to *Francisella* by disrupting inflammasome activation and NK cells", Cornut and colleagues investigate the role of interferon-regulatory factor IRF2 in macrophage and NK cell functions during inflammasome activation caused by *Francisella novicida* infection. In macrophages, the authors confirmed that IRF2 is an essential component driving IL-1b and IL-18 secretion, owing to the role of IRF2 as a transcriptional regulator of gasdermin D. They also showed that IRF2-KO mice are more susceptible to *Francisella* infection and had less IFN-g owing to reduced NK cell number and maturation. Finally, administration of IFN-g protects IRF2-KO mice against *Francisella* infection.

This work nicely follows on two previous studies from the same lab and another lab demonstrating that IRF2 is an important contributor of inflammasome activation. This manuscript convincingly offers new physiological insights into IRF2 and inflammasome activation in response to a bacterial pathogen, which had not been demonstrated previously, and further links this host response to NK cells.

Major comments:

1. A strength of this paper is the generation of a new IRF2-KO mouse strain. However, no data is provided to verify that the KO is a KO. Please provide sequencing data of the truncated IRF2 gene and protein data showing the lack of IRF2. It would also be useful to verify the expression of IRF1 in this new mouse strain due to the compensatory role between IRF1 and IRF2 in inflammasome activation.

2. A few observations in this manuscript should be explored further: (A) It is unusual to see a role for caspase-11 for IL-1b and IL-18 release during *F. novicida* infection (Figure 2C and D). How is this finding explained given that the tetra-acylated lipid A of *F. novicida* LPS cannot activate caspase-11? (B) IRF2 appears to have a negative regulatory role in TNF production in BMDMs (Figure 2E). Can the authors check that the activation of the NF-kB pathways is similar between WT and IRF2-KO BMDMs? (C)

Given that the authors have found a defect in IFN-g production in IRF2-KO mice, it would also be important to check IFN-b levels since this important cytokine also primes the inflammasome during Francisella infection in mice and BMDMs.

3. An interesting observation arising from this manuscript is that IRF2, via its regulatory role in GSDMD expression, controls the secretion of IL-18 in BMDMs, whereas the *in vivo* role for IRF2 in IL-18 secretion is less clear. During *F. novicida* infection, IRF2-KO mice had more IL-18 than WT mice and this appears to be a result of the increased bacterial burden in the IRF2-KO mice. Since the authors have also established the *E. coli* OMV system shown in Fig S2, I suggest injecting WT, IRF2-KO, and GSDMD-KO mice with *E. coli* OMV and then monitor for IL-18 secretion. In this way, the IRF2-GSDMD-IL-18 axis can properly be investigated *in vivo*, with IL-18 levels not being directly affected by differences in bacterial proliferation/burden.

Minor comments:

1 It remains unclear whether the IFN-g administration increased the maturation of NK cells. Please provide data to show that NK cell maturation is indeed rescued to add weight to the NK cell numbers in Figure 6E-G.

2 Figure 1A survival data only represents one experiment instead of pooling from the available data from three independent experiments. This is an important claim in the paper and the data supporting this claim should be pooled from all three independent repeats to increase the robustness of the claim.

3 Please indicate in the figure legends the number of independent biological repeats that were performed. For example, I cannot tell whether any of the *in vivo* challenge experiments were done one or more times. In particular, Figure 5A and 6A appear to have been done only once. Please provide pooled data from at least two independent biological repeats.

4 Several references were duplicated. e.g. Benaoudia et al 2019; Kayagaki et al 2019; Thygesen et al 2019. Please double check the reference list.

Referee #2:

This manuscript by Cornut et al. assessed the effect of IRF2 deficiency on inflammasome activation and NK cell maturation during *Francisella novicida* infection in mice.

One conclusion is that IRF2 deficiency inhibits NK cell maturation. While this is not a novel finding, this study provides additional information including reduced levels of NK cell effector molecules and tries to position where IRF2 functions in the pathway of NK cell maturation.

A second conclusion is that IRF2 deficiency disrupts inflammasome function. This is true for the *in vitro* studies with bone marrow derived macrophages (BMDMs) and confirms previous *in vitro* studies. However, the result *in vivo* is not clear cut as some GSDMD is detected, and IL-1b, IL-18, and cleaved gasdermin D which are outputs of inflammasome activation, are elevated above the wild type mouse levels. This may be largely or partially due to the more severe infection in the knockout. Overall conclusions about inflammasome function *in vivo* remain somewhat unclear.

Finally, IFN-g therapy was shown to prolong survival of IRF2 deficient mice and this was a nice end to the study, although it should be noted that treatment was from the time of infection, so this boosted macrophage activation to provide immediate enhanced protection from infection.

The experimental approach is generally clear and the data is of excellent quality and well presented. However, I think that publication in EMBO Reports will rely on an editorial decision on novelty and interest. The effects of IRF2 knockout on GSDMD expression and NK cell development are already established in the literature. It is not surprising that knockout of a transcription factor can have effects on a number of systems. The authors need to work to convince the editors of the novelty of the work and the importance of the findings, although the quality of results is not in doubt.

Specific comments

1. The title needs some more thought. I don't think the data from this paper supports "disrupting inflammasome activation" as a cause of reduced resistance to *Francisella*. Although the effect of GSDMD ko confirms its important role in resistance the IRF2 ko was not completely lacking GSDMD or inflammasome function. IL-18 and IL-1b were more elevated *in vivo* than in wild type - this may reflect the higher bacterial load. However, inflammasome activation *per se* is not disrupted even though the rapid release of IL-18 and IL-1b may be. *In vitro* work showed elevated cleavage of caspase-1, so the inflammasome machinery other than GSDMD seems functional. Also it is not precise to write "disrupting NK cells" as that might mean lysing them - but you could be could write "disrupting NK cell maturation". Maybe something like "IRF2 deficiency disrupts pyroptosis, NK cell interferon-g production and resistance to *Francisella*."

2. Prior work has shown that the AIM2 inflammasome is essential for efficient control of *Francisella*. Is there any information on what aspects of inflammasome function are most important? Are there IL-1b or IL-18 pathway knockouts that have been tested

as well as GSDMD ko done here? Given the elevated IL-18 in the IRF2 ko mouse I am wondering whether efficient pyroptosis is really the main requirement for control.

3. The introduction needs some refinement; paragraph 2 is very long. The function of Gasdermin D is not explained.

4. Line 399. Can data in Figure 1F and H be compared to say that infection induced GSDMD mRNA, or did you never compare directly?

5. I think that data on IL-1beta, such as is in Figure S1B should be in the main figures not supplementary since it shows the same trend as IL-18.

6. IRF2 is clearly required for all GSDMD expression in macrophages in Figure 2 but GSDMD is detectable in vivo in the IRF2 ko. Later in the paper you note that IRF1 can substitute for IRF2. However, I think alternative promoter usage is worth considering to explain the expression of GSDMD in vivo. I notice that the Riken Zenbu FANTOM5 mouse promoterome that maps transcription starts shows that macrophages always use the main promoter for non-coding exon 1, but there is some transcription particularly in gut epithelium, starting just upstream of exon 2. Maybe some spleen cells are using an alternative promoter.

7. Fig 2D and Fig S2A. I had presumed that IL-18 release at 24 hours in GSDMD and IRF2 ko likely reflected eventual membrane breakdown in cells undergoing inflammasome activation without GSDMD. But I note that in Figure S2A IL-1b is released at 24 hours of E. coli or OMV treatment by only IRF2 ko cells and not by GSDMD cells. Did you look at IL-1b after 24 hours of Francisella infection, to see if its release parallels IL-18 in Fig 2D?

8. Regarding Fig S2A can you be more specific about what you mean by "partial complementation" in Line 441-442? What kind of mechanism do you envisage? Do you think that GSDMD is being expressed?

9. Line 444-445 Maybe it is worth commenting that TNF is higher in the IRF2 ko in Figure 2E - why do you think this is? This is particularly in the FPI mutant.

10. Line 465-466 You state that IL-15 had no effect on IFN γ levels, but maybe it should say "IL-15 alone". It seemed to have a clear effect when combined with IL-18 on the IRF2 ko BMDM culture with SMC. Should this not be noted? Or is this what the result would look like without the BMDM present at all - i.e. just the effect of the cytokines on the SMC? I am having trouble interpreting Figure 3. Don't we need to see the effect of IL-15 and IL-18 on SMC in the absence of BMDM to interpret these results? Also I think the text needs more information on regulation of IFN γ release by spleen mononuclear cells - e.g. by IL-12, 15 and 18 or other cytokines? for us to understand this result.

11. Line 531-532 "Eomes expression was slightly reduced in the lung" - the data cannot give certainty on this. It is interesting that changes in both the transcription factor expression and GzmA and IL-18R expression are modest in lung and much stronger in spleen. Any ideas on this?

12. Figure 5J- we can't really see that Ki67 expression is the same in all genotypes, as this style of plot obscures two of the results. It actually look like there is much less Ki67 staining in WT and GSDMD ko. I think normal overlaid plots with different lines/shading would be better.

Minor Points:

1. line 48-49 in the abstract: "These findings establish IRF2 as a dual regulator of inflammasome activity and NK cell function". Confirm would be more appropriate than establish, since both these roles for IRF2 are already published.

2. Line 70 strain not strains.

3. Line 71 - "so called" is not necessary

4. Line 87 - delete the comma

5. Line 388 - Figure 1C is not appropriate for that sentence - probably for the previous sentence. However, the statement that IL-12 and IL-15 are involved in IFN γ production could do with a reference.

6. Isn't figure S2D just the first two samples from Figure S2C?

7. Line 461 - please insert infected "at MOI1 and 10" to indicate the difference between the difference between the supp data and the main figure. Took me a while to work out.

8. Line 510 Figure 4E-F The text says that other immune cell types were present in normal numbers. Just say ILC subsets rather than other immune cell types, because you haven't shown most cell types. You might need a split axis to actually see anything useful about the ILC numbers in this figure.

9. The writing is too small on some figures. Please check all figures for legibility.

10. Line 515-516 Perhaps it makes more sense to suggest that loss of IRF2 slows the release of NK cells from the bone marrow, leading to their increased number, rather than suggesting that it regulates NK cells at later stages, since in the next sentence you show profound differences in the bone marrow NK phenotype.

11. In general where there are trends that are not statistically significant, language needs to be modified to be less definite about the result.

12. Some references in your reference list are duplicated.

13. Figure legends need some editing to ensure things are written conventionally. Line 745-746; It is conventional to start with a description of the panels in alphabetic order.

14. Materials and Methods needs a bit of formatting eg some sections are written as multiple short paragraphs.

15. Line 634-635. Mariathasan should not be cited to say that GSDMD regulates IL-18 levels since the paper is from 2005 and they could not have tested the role of GSDMD.

16. Line 379-378; "IFN- γ levels were decreased in the blood of IRF2 ko mice, pointing to a critical role of IRF2 in regulating IFN- γ " Since the role of IRF2 was later found to be quite indirect in regulating NK maturation, this might be better phrased.

Referee #3:

IRF2 is a transcription factor that has been previously shown to have a role in inflammasome activation through its regulation of caspase-4 and GSDMD levels. In this manuscript, the authors set out to understand the role of the transcription factor IRF2 in regulating in vivo responses to infection. The authors generated IRF2-deficient mice and find that they are as susceptible to *Francisella novicida* as GSDMD-deficient mice. They also find that IRF2-deficient mice also had a defect in IFN γ production, possibly due to a cell intrinsic role for IRF2 in regulating the number and maturation of NK cells. IRF2-deficient NK cells had a defect in expression of IL-18R and granzyme A. They were able to partially restore immune responses and control of infection by treatment of IRF2-deficient mice with recombinant IFN γ . Overall, the study is carefully designed, thorough, and well-written. These findings provide new insight into the role of IRF2 in regulating inflammasome activation as well as NK cell function. Furthermore, these findings provide a foundation for the potential use of IFN γ therapy in treating bacterial infections in patients with primary immunodeficiencies. This study will be of broad interest to basic researchers and clinicians in the fields of innate immunity, primary immunodeficiencies, and bacterial pathogenesis.

Minor comments:

1. Fig. 1H: are there statistically significant differences in GSDMD transcript levels between WT and IRF2 ko mice? If not, the authors should reword the description of this data, as it currently says in the text that "We demonstrated that IRF2 positively regulated GSDMD transcript 398 and protein levels, both in naive (Figure 1F-G) and infected mice (Figure 1H-I, Figure 399 S1C)." In addition, the authors may want to consider using ImageJ to quantify the protein levels of GSDMD in WT and IRF2 ko for figures 1G and I to determine whether there are indeed differences in GSDMD protein levels between WT and IRF2 ko mice.
2. In some of the figures, such as Figure 5G-J, the authors show flow cytometry plots of histograms that they say is concatenate of 2 mice. Is it possible for the authors to do at least one more mouse for each of these analyses and then obtain statistical analysis?

EMBOR-2025-61421V1

Cornut et al: IRF2 deficiency weakens resistance to Francisella by disrupting inflammasome activation and NK cells

Response to Reviewer's Comments

Reviewer #1:

General comment: *In this manuscript "IRF2 deficiency weakens resistance to Francisella by disrupting inflammasome activation and NK cells", Cornut and colleagues investigate the role of interferon-regulatory factor IRF2 in macrophage and NK cell functions during inflammasome activation caused by Francisella novicida infection. In macrophages, the authors confirmed that IRF2 is an essential component driving IL-1 β and IL-18 secretion, owing to the role of IRF2 as a transcriptional regulator of gasdermin D. They also showed that IRF2-KO mice are more susceptible to Francisella infection and had less IFN- γ owing to reduced NK cell number and maturation. Finally, administration of IFN- γ protects IRF2-KO mice against Francisella infection.*

This work nicely follows on two previous studies from the same lab and another lab demonstrating that IRF2 is an important contributor of inflammasome activation. This manuscript convincingly offers new physiological insights into IRF2 and inflammasome activation in response to a bacterial pathogen, which had not been demonstrated previously, and further links this host response to NK cells.

We thank reviewer 1 for her/his appraisal of our study, and her/his constructive comments.

Major comments:

1. *A strength of this paper is the generation of a new IRF2-KO mouse strain. However, no data is provided to verify that the KO is a KO. Please provide sequencing data of the truncated IRF2 gene and protein data showing the lack of IRF2. It would also be useful to verify the expression of IRF1 in this new mouse strain due to the compensatory role between IRF1 and IRF2 in inflammasome activation.*

IRF2^{KO} mice were generated using the CRISPR-Cas9 genome editing system, as previously described (Teixeira et al, 2018). The reviewer is correct that this new mouse model was not sufficiently detailed in the original version of the manuscript. To address this issue, we have now provided a schematic representation of the targeted deletion (new Figure S1A), an example of genotyping showing WT, heterozygous and homozygous mice (new Figure S1B) and confirmation of IRF2 loss in BMDMs by immunoblotting (new Figure S1C). In addition, we examined IRF1 expression as a potential compensatory mechanism following IRF2 deficiency, assessing both protein and transcript levels in spleen and BMDMs. We did not observe any IRF1 increase in IRF2^{KO} mice ruling out a compensatory mechanism (new Figures S1D-F).

In response to the reviewer's comment, the following text has been included in the revised manuscript:

To explore the *in vivo* role of IRF2 in the immune responses against *F. novicida*, we generated IRF2^{KO} mice using CRISPR/Cas9 genome editing (Figure S1A-B). Successful *IRF2* deletion was confirmed by immunoblotting (Figure S1C), while IRF1 protein levels remained unchanged in the absence of *IRF2* (Figure S1D-F). These findings confirm that IRF2 deficiency does not trigger compensatory IRF1 expression, supporting the conclusion that the observed phenotypes are directly attributable to IRF2 loss.

2. A few observations in this manuscript should be explored further: (A) It is unusual to see a role for caspase-11 for IL-1 β and IL-18 release during *F. novicida* infection (Figure 2C and D). How is this finding explained given that the tetra-acylated lipid A of *F. novicida* LPS cannot activate caspase-11? (B) IRF2 appears to have a negative regulatory role in TNF production in BMDMs (Figure 2E). Can the authors check that the activation of the NF- κ B pathways is similar between WT and IRF2-KO BMDMs? (C) Given that the authors have found a defect in IFN- γ production in IRF2-KO mice, it would also be important to check IFN- β levels since this important cytokine also primes the inflammasome during *Francisella* infection in mice and BMDM

(A) It is true that the atypical tetra-acylated lipid A structure of *F. novicida* LPS does not efficiently activate caspase-11 under basal conditions. In this study, we observed a minor contribution of caspase-11 (compare Casp11^{KO} to AIM2^{KO}) during *F. novicida* infection possibly due to the heterogeneity of LPS in whole bacteria and to the presence of a minor fraction of penta-acylated LPS.

The following sentences have been added in the revised manuscript:

"A minor contribution of caspase-11 in IL-1 β and IL-18 was observed especially at high MOI. This observation was unexpected due to the inability of *F. novicida* tetra-acylated LPS (Hagar et al, 2013; Lagrange et al, 2018) to activate caspase-11 and suggests that LPS molecules with higher acylation levels might be present at low frequency in *F. novicida*. »

(B) While we observed some variability between WT and IRF2^{KO} samples, these differences were inconsistent and lacked reproducibility. To further clarify this point, we conducted additional experiments, the results of which are shown below. Unless the editor or reviewers strongly consider that this aspect warrants deeper investigation, we prefer not to complicate the main manuscript with data that may not substantially impact the central conclusions.

Nonetheless, we acknowledge that those observations are of potential interest.

The following sentence has been added to the revised manuscript.

"Some variability was observed between WT and IRF2^{KO} samples, but these differences were inconsistent and lacked reproducibility"

Figure 1: *BMDMs from all genotypes produced comparable levels of TNF in response to F. novicida infection*

BMDMs from the indicated genotypes were infected or not with *F. novicida* (U112) at a MOI of 10 or 100, or with its avirulent mutant Δ FPI. TNF concentration was determined by ELISA 6 hours p.i. Each dot corresponds to one replicate; the bars indicate the mean and the standard error of the mean (SEM) from data pooled from three independent experiments.

(C) An increase in IFN- β levels was observed in IRF2^{KO} mice compared to WT mice following *Francisella* infection (new Figure 1G). Interestingly, this increase in IFN- β contrasts with IFN- γ levels, which were reduced in IRF2^{KO} mice relative to WT mice. Importantly, since IFN- β levels were not diminished in IRF2^{KO} mice, the impaired inflammasome activation observed in these animals cannot be attributed to a deficiency in type I interferon responses.

The following sentence has been added in the revised manuscript:

"IFN- β levels were significantly increased in IRF2^{KO} mice compared to WT mice following Francisella infection (Figure 1G)".

3. An interesting observation arising from this manuscript is that IRF2, via its regulatory role in GSDMD expression, controls the secretion of IL-18 in BMDMs, whereas the *in vivo* role for IRF2 in IL-18 secretion is less clear. During *F. novicida* infection, IRF2-KO mice had more IL-18 than WT mice and this appears to be a result of the increased bacterial burden in the IRF2-KO mice. Since the authors have also established the *E. coli* OMV system shown in Fig S2, I suggest injecting WT, IRF2-KO, and GSDMD-KO mice with *E. coli* OMV and then monitor for IL-18 secretion. In this way, the IRF2-GSDMD-IL-18 axis can properly be investigated *in vivo*, with IL-18 levels not being directly affected by differences in bacterial proliferation/burden.

We thank the reviewer for this insightful comment.

The standard protocol recommends injecting first poly(I:C) then outer membrane vesicles (OMVs) before monitoring cytokines *in vivo* (Vanaja et al., 2016). Since poly(I:C), as a strong IFN inducer, may partially complement IRF2 deficiency, we instead used a single lipopolysaccharide (LPS) injection.

As shown in new Figure 5E of the revised manuscript, IL-18 levels were reduced in both IRF2^{KO} and GSDMD^{KO} mice following LPS challenge, a result that contrasts with the elevated IL-18 levels observed in IRF2^{KO} mice during *Francisella* infection *in vivo*. These findings support the conclusion that elevated IL-18 levels in IRF2-deficient (and GSDMD^{KO}) mice during infection are a downstream consequence of uncontrolled bacterial replication, rather than a direct effect of IRF2 loss on inflammasome signaling.

This clarification has now been incorporated into the manuscript. *"The increased IL-18 levels observed in both IRF2^{KO} and GSDMD^{KO} mice may be due to the increased bacterial burden observed in these two mice (Figure 5B). To avoid this bacterial burden-associated bias, we injected mice with LPS at 3 µg per gram of body weight for 6 hours. IL-18 levels were reduced in both IRF2^{KO} and GSDMD^{KO} mice compared to WT mice (Figure 5E). These results demonstrate that in vivo, in the absence of confounding factors, IRF2 (and GSDMD) positively controls IL-18 levels"*.

Minor comments:

1 *It remains unclear whether the IFN-γ administration increased the maturation of NK cells. Please provide data to show that NK cell maturation is indeed rescued to add weight to the NK cell numbers in Figure 6E-G.*

Following IFN-γ treatment, NK cell numbers increased in both WT and IRF2^{KO} mice. Interestingly, despite this expansion, the cells maintained an immature phenotype (new Figures 6F and 6I) but upregulated granzyme A expression (new Figures 6G and 6J).

The following text has been added in the revised manuscript:

"Interestingly, despite this expansion, the cells maintained an immature phenotype (Figure 6F, blood and 6I, spleen). Notably, granzyme A expression showed a trend toward upregulation (Figure 6G, blood and 6J, spleen), suggesting that, in the presence of IFN-γ, IRF2-deficient NK cells may retain some cytotoxic potential despite their immature status."

2 *Figure 1A survival data only represents one experiment instead of pooling from the available data from three independent experiments. This is an important claim in the paper and the data supporting this claim should be pooled from all three independent repeats to increase the robustness of the claim.*

Data from three independent experiments were pooled, still demonstrating a statistically significant difference in survival between infected WT and IRF2^{KO} mice (Figure 1A) with a total of 20 WT and 17 IRF2^{KO} mice.

3 *Please indicate in the figure legends the number of independent biological repeats that were performed. For example, I cannot tell whether any of the in vivo challenge*

experiments were done one or more times. In particular, Figure 5A and 6A appear to have been done only once. Please provide pooled data from at least two independent biological repeats.

Experiments have been performed at least three times, except for Figures 5A and 6A, as the reviewer pointed out. For these, we conducted new independent experiments and pooled the data. The results show a significant difference in survival between WT and IRF2^{KO} or GSDMD^{KO} mice (Figure 5A), as well as between IRF2^{KO} mice with or without IFN- γ treatment, and GSDMD^{KO} mice with or without IFN- γ treatment (Figure 6A).

We have checked the figure legends to ensure that the numbers of independent experiments are indicated.

4 Several references were duplicated. e.g. Benaoudia et al 2019; Kayagaki et al 2019; Thygesen et al 2019. Please double check the reference list.

Duplicate references have been removed.

Referee #2:

General comment: *This manuscript by Cornut et al. assessed the effect of IRF2 deficiency on inflammasome activation and NK cell maturation during Francisella novicida infection in mice.*

One conclusion is that IRF2 deficiency inhibits NK cell maturation. While this is not a novel finding, this study provides additional information including reduced levels of NK cell effector molecules and tries to position where IRF2 functions in the pathway of NK cell maturation.

A second conclusion is that IRF2 deficiency disrupts inflammasome function. This is true for the in vitro studies with bone marrow derived macrophages (BMDMs) and confirms previous in vitro studies. However, the result in vivo is not clear cut as some GSDMD is detected, and IL-1b, IL-18, and cleaved gasdermin D which are outputs of inflammasome activation, are elevated above the wild type mouse levels. This may be largely or partially due to the more severe infection in the knockout. Overall conclusions about inflammasome function in vivo remain somewhat unclear.

Finally, IFN-g therapy was shown to prolong survival of IRF2 deficient mice and this was a nice end to the study, although it should be noted that treatment was from the time of infection, so this boosted macrophage activation to provide immediate enhanced protection from infection.

The experimental approach is generally clear and the data is of excellent quality and well presented. However, I think that publication in EMBO Reports will rely on an editorial decision on novelty and interest. The effects of IRF2 knockout on GSDMD expression and NK cell development are already established in the literature. It is not surprising that knockout of a transcription factor can have effects on a number of systems. The authors need to work to convince the editors of the novelty of the work and the importance of the findings, although the quality of results is not in doubt.

We thank reviewer 2 for her/his appraisal of our study, and her/his constructive feedback.

We appreciate this insightful comment. Our study focuses on the novel *in vivo* regulation of GSDMD by IRF2 in both naive and infected mice, which, to our knowledge, has not been previously demonstrated. Given the central role of GSDMD in numerous inflammatory processes, we believe this represents an important contribution to the field.

In addition, while we agree that the cell intrinsic role of IRF2 in NK cell maturation has been previously reported, our manuscript extends these findings by demonstrating this effect and its consequences within an integrated infection model. We further show that IRF2 influences the antibacterial effectors of NK cells, including GzmA and IL-18R expression.

Finally, we provide evidence that IFN- γ supplementation can restore both GSDMD expression and NK cell numbers, resulting in predominantly immature NK cells that nevertheless exhibit increased GzmA expression and contribute to an enhanced

immune response against *F. novicida*. Altogether, we believe our study provides new mechanistic insights and will be of interest for the readers of EMBO reports.

Specific comments

1. The title needs some more thought. I don't think the data from this paper supports "disrupting inflammasome activation" as a cause of reduced resistance to *Francisella*. Although the effect of GSDMD ko confirms is important role in resistance the IRF2 ko was not completely lacking GSDMD or inflammasome function. IL-18 and IL-1 β were more elevated in vivo than in wild type - this may reflect the higher bacterial load. However, inflammasome activation per se is not disrupted even though the rapid release of IL-18 and IL-1 β may be. In vitro work showed elevated cleavage of caspase-1, so the inflammasome machinery other than GSDMD seems functional. Also it is not precise to write "disrupting NK cells" as that might mean lysing them - but you could be could write "disrupting NK cell maturation". Maybe something like "IRF2 deficiency disrupts pyroptosis, NK cell interferon- γ production and resistance to *Francisella*."

We thank reviewer 2 for her/his suggestion and have followed her/his advice.

The revised title is: **IRF2 deficiency disrupts pyroptosis, NK cell interferon- γ production and resistance to *Francisella***

2. Prior work has shown that the AIM2 inflammasome is essential for efficient control of *Francisella*. Is there any information on what aspects of inflammasome function are most important? Are there IL-1 β or IL-18 pathway knockouts that have been tested as well as GSDMD ko done here? Given the elevated IL-18 in the IRF2 ko mouse I am wondering whether efficient pyroptosis is really the main requirement for control.

Previous studies have investigated the roles of IL-1 β and IL-18 in host responses to *Francisella* infection and demonstrated that both cytokines contribute to host defense, albeit only partially (see figure 4I from (Mariathasan *et al*, 2005) below). These findings suggest that additional, cytokine-independent mechanisms also play a role (likely pyroptosis) (Mariathasan *et al*, 2005; Zhu *et al*, 2018). Side by side comparison of IL1R/IL18R^{DKO}, AIM2^{KO} and GSDMD^{KO} suggest that IL-1 β and IL-18 have a greater contribution than GSDMD to the innate immune response to *Francisella* (Zhu *et al*, 2018-see figure 1A from this paper below). Collectively, IL-1 β , IL-18, and pyroptosis each contribute to the overall defense against *Francisella*. In vitro, Aim2^{KO} and ASC^{KO} have a stronger inflammasome deficit than caspase-1/11^{DKO} and GSDMD^{KO} (Pierini *et al*, 2012; Zhu *et al*, 2018).

Figure 2: Immune responses to *F. novicida* depend on both cytokines-dependent and independent mechanisms with IL-1/IL-18 signaling playing a greater role than GSDMD.

Left panel: figure 4I from (Mariathasan *et al*, 2005). WT mice were injected with either isotype controls or IL-1 β /IL-18 neutralizing antibodies and infected with *F. novicida*. The bacterial burden was monitored at 48h post-inoculation. Right panel : figure 1A from (Zhu *et al*, 2018) showing survival curves of the indicated KO mice in response to *F. novicida* infection.

The following sentence has been added: «Cytokines and GSDMD-dependent responses contribute to host defense to *F. novicida* although side-by-side comparisons of IL-1R/IL-18^{DKO} mice and GSDMD^{KO} mice suggest that cytokines are more important than GSDMD to promote the survival of mice to *F. novicida* infection (Zhu *et al.*, 2018; Mariathasan *et al.*, 2005)».

3. The introduction needs some refinement; paragraph 2 is very long. The function of Gasdermin D is not explained.

The introduction has been revised in line with your suggestions.

“*F. tularensis* ssp. *novicida*, also known as *F. novicida* (Johansson *et al.*, 2010), is a bacterium that is avirulent for immunocompetent humans but is highly virulent in mouse models of tularemia. Immune responses to *F. novicida* are dependent on the AIM2 inflammasome (Fernandes-Alnemri *et al.*, 2010; Belhocine and Monack, 2012; Meunier *et al.*, 2015). Activated caspase-1 within the AIM2 inflammasome cleaves pro-IL-1 β and pro-IL-18, generating their active forms IL-1 β and IL-18. Caspase-1 also cleaves Gasdermin D (GSDMD), producing a N-terminal fragment that forms pores in the host cell membrane. These GSDMD pores enable the release of mature IL-1 β and IL-18 and drive pyroptosis, an inflammatory form of cell death that restricts bacterial replication (Zhu *et al.*, 2018; Shi *et al.*, 2015; He *et al.*, 2015). IL-18 promotes IFN- γ production by NK cells during the early phase of *F. novicida* infection (Pierini *et al.*, 2013). Cytokines and GSDMD-dependent responses contribute to host defense to *F. novicida* although side-by-side comparisons of IL-1R/IL-18^{DKO} mice and GSDMD^{KO} mice suggest that cytokines are more important than GSDMD to promote the survival of mice to *F. novicida* infection (Zhu *et al.*, 2018; Mariathasan *et al.*, 2005). Alongside this canonical, caspase-1-dependent inflammasome pathway, the non-canonical caspase-11-dependent inflammasome (and its human orthologs, caspase-4 and -5) directly sense cytosolic lipopolysaccharide (LPS) and cleave GSDMD to trigger pyroptosis (Shi *et al.*, 2015; Kayagaki *et al.*, 2015; Yang *et al.*, 2015; Vak *et al.*, 2019). *F. novicida*, due to its tetra-acylated LPS, escapes caspase-11 detection (Hagar *et al.*, 2013)».

4. Line 399. Can data in Figure 1F and H be compared to say that infection induced GSDMD mRNA, or did you never compare directly?

We now provide a direct comparison of GSDMD transcript levels in uninfected and infected mice demonstrating that indeed, *Francisella* infection induced GSDMD expression (new Figure S1G of the revised manuscript).

The following sentence has been added:

“Of note, GSDMD transcript levels were induced (3- to 4-fold increase depending on the organ) upon infection, with a statistically significant change observed only in the liver (Figure S1G).”

5. I think that data on IL-1beta, such as is in Figure S1B should be in the main figures not supplementary since it shows the same trend as IL-18.

This Figure is now new Figure 1D.

6. IRF2 is clearly required for all GSDMD expression in macrophages in Figure 2 but GSDMD is detectable *in vivo* in the IRF2 ko. Later in the paper you note that IRF1 can substitute for IRF2. However, I think alternative promoter usage is worth considering to explain the expression of GSDMD *in vivo*. I notice that the Riken Zenbu FANTOM5 mouse promoterome that maps transcription starts shows that macrophages always use the main promoter for non-coding exon 1, but there is some transcription particularly in gut epithelium, starting just upstream of exon 2. Maybe some spleen cells are using an alternative promoter.

The low yet detectable levels of GSDMD expression observed *in vivo* in IRF2^{KO} mice may result from compensatory activity by IRF1. Alternatively, this residual GSDMD expression could indeed be explained by the presence of an alternative promoter, in splenic cells. Since exon 1 is non-coding and the proposed alternative transcription start site (TSS) lies upstream of exon 2, no change in protein size would be expected. Although this hypothesis is interesting, testing it would require technically demanding experiments, such as precise mapping of GSDMD mRNA isoforms to confirm alternative TSS usage. This possibility has been acknowledged in the text: "..., despite a large reduction in full-length GSDMD levels, the cleaved GSDMD fragment was specifically observed in the spleen from infected IRF2^{KO} mice ..., possibly due to ... or the potential contribution of an alternative IRF2-independent promoter that may be active in splenic cells."

7. Fig 2D and Fig S2A. I had presumed that IL-18 release at 24 hours in GSDMD and IRF2 ko likely reflected eventual membrane breakdown in cells undergoing inflammasome activation without GSDMD. But I note that in Figure S2A IL-1b is released at 24 hours of *E. coli* or OMV treatment by only IRF2 ko cells and not by GSDMD cells. Did you look at IL-1b after 24 hours of *Francisella* infection, to see if its release parallels IL-18 in Fig 2D?

After 24 hours of *Francisella* infection, IL-1 β was released at comparable levels in WT, IRF2^{KO}, GSDMD^{KO}, and CASP 11^{KO} cells, but was still fully dependent on ASC (new Figure S2A), indicating differences in IL-1 β and IL-18 release modalities even at this late time point. As expected at 6 h p.i., the response is fully IRF2- and GSDMD-dependent (see Figure below).

Figure 3: IL-1 β release is fully IRF2- and GSDMD-dependent at 6 hours p.i., even though this is not the case at 24 hours p.i. BMDMs from the indicated genotypes were infected or not with *F. novicida* (U112) at a MOI of 10 or 100, or with its avirulent mutant Δ FPI. IL-1 β concentration was determined by ELISA 6 hours p.i. Each dot corresponds to one replicate; the bar shows the mean and the standard error of the mean (SEM) of 3 replicates from one experiment. Kruskal-Wallis analysis with Dunn's correction was performed.

The following text has been added in the result section:

Interestingly, at 24 h p.i., IL-18 levels increased in the supernatants of infected IRF2^{KO} and GSDMD^{KO} cells, albeit to a much lower extent than in WT macrophages, while IL-18 remained low to undetectable in AIM2^{KO} and ASC^{KO} BMDMs (Figure 2D, right panel). At the same 24-hour time point, IL-1 β release was still fully dependent on ASC, but measured at comparable levels in WT, IRF2^{KO}, GSDMD^{KO}, and CASP11^{KO} cells (Figure S2A). These findings suggest that although both IL-1 β and IL-18 are regulated by IRF2 and GSDMD, their secretion follows distinct temporal patterns during *Francisella* infection. In particular, at late time points of infection, IL-1 β release appears to be less dependent on the IRF2-GSDMD axis than IL-18 release.

8. Regarding Fig S2A can you be more specific about what you mean by "partial complementation" in Line 441-442? What kind of mechanism do you envisage? Do you think that GSDMD is being expressed?

This comment is linked to the Figure S2B of the revised manuscript and to the fact that in the presence of *E. coli* or OMVs for 24h, IL-1 β release is only partially dependent on IRF2 while being fully dependent on GSDMD.

We envisage that type I IFN induced by *E. coli* LPS/OMVs may induce GSDMD in an IRF2-independent manner (presumably in an IRF-1-dependent manner).

This line has been added to the Results section: "...suggesting that type I IFN produced during the 24-hour treatment period in response to *E. coli* LPS, may partially restore inflammasome activity, likely through the IRF2-independent induction of GSDMD expression".

9. Line 444-445 Maybe it is worth commenting that TNF is higher in the IRF2 ko in Figure 2E - why do you think this is? This is particularly in the FPI mutant.

Reviewer #1 raised a similar and insightful question, and we provided the following response:

While we observed some variability between WT and IRF2^{KO} samples, these differences were inconsistent and lacked reproducibility. To further clarify this point, we conducted additional experiments, the results of which are shown below. Unless the editor or reviewers strongly consider that this aspect warrants deeper investigation, we prefer not to complicate the main manuscript with data that may not substantially impact the central conclusions.

Nonetheless, we acknowledge that those observations are of potential interest.

The following sentence has been added to the revised manuscript.

“Some variability was observed between WT and IRF2^{KO} samples, but these differences were inconsistent and lacked reproducibility”

Figure 4 (same as Figure 1):

BMDMs from all genotypes produced comparable levels of TNF in response to *F. novicida* infection

BMDMs from the indicated genotypes were infected or not with *F. novicida* (U112) at a MOI of 10 or 100, or with its avirulent mutant ΔFPI. TNF concentration was determined by ELISA 6 hours p.i. Each dot corresponds to one replicate; the bars indicate the mean and the standard error of the mean (SEM) from data pooled from three independent experiments.

10. Line 465-466 You state that IL-15 had no effect on IFN γ levels, but maybe it should say "IL-15 alone". It seemed to have a clear effect when combined with IL-18

on the IRF2 ko BMDM culture with SMC. Should this not be noted? Or is this what the result would look like without the BMDM present at all - i.e. just the effect of the cytokines on the SMC? I am having trouble interpreting Figure 3. Don't we need to see the effect of IL-15 and IL-18 on SMC in the absence of BMDM to interpret these results? Also I think the text needs more information on regulation of IFN γ release by spleen mononuclear cells - e.g. by IL-12, 15 and 18 or other cytokines? for us to understand this result.

We thank the reviewer for this helpful suggestion. In response, we made several clarifications and additions in the manuscript:

- 1) We added the sentence: "Previous studies have shown that IFN- γ production by splenic mononuclear cells (SMCs) can be stimulated by combinations of cytokines such as IL-12, IL-15, and IL-18 (Nielsen et al., 2016; Otani et al, 1999)."
- 2) We explicitly included the "IL-15 alone" condition in the revised description and emphasized that "a significant increase in IFN- γ levels was observed only when IL-15 was combined with IL-18". This supports the interpretation that IRF2 regulates IFN- γ production primarily via its control of GSDMD expression and subsequent IL-18 release from macrophages, rather than through IL-15 signaling alone.
- 3) We also emphasized the controls involving cytokine effects in the absence of BMDMs to ensure clarity. As already shown in the manuscript, these conditions demonstrate no IFN- γ production. We highlighted this more clearly in the revised text with the sentence: "As expected, no IFN- γ was produced either in the absence of BMDM infection or in the absence of SMCs (Figure S2D-MOI1; and Figure S2E-MOI10)."

11. Line 531-532 "Eomes expression was slightly reduced in the lung" - the data cannot give certainty on this. It is interesting that changes in both the transcription factor expression and GzmA and IL-18R expression are modest in lung and much stronger in spleen. Any ideas on this?

Differences in the pulmonary microbiota may play a role in the distinct immune regulation observed between the lung and spleen. However, the mechanisms underlying these organ-specific responses remain poorly understood and require further investigation.

12. Figure 5J- we can't really see that Ki67 expression is the same in all genotypes, as this style of plot obscures two of the results. It actually look like there is much less Ki67 staining in WT and GSDMD ko. I think normal overlaid plots with different lines/shading would be better.

This observation is accurate: Ki67 staining is reduced in both WT and GSDMD^{KO} cells. The plots and the text have been updated accordingly to better highlight these differences (new Figure 5K).

The following sentence has been added to the text:

"NK cell proliferation, as assessed by the Ki67 assay, was significantly higher in IRF2^{KO} mice, compared to both WT and GSDMD^{KO} mice, indicating increased turnover (Figure 5K)".

Minor Points:

1. *line 48-49 in the abstract: "These findings establish IRF2 as a dual regulator of inflammasome activity and NK cell function". Confirm would be more appropriate than establish, since both these roles for IRF2 are already published.*

The change has been made:

"These findings **confirm** IRF2 as a dual regulator of inflammasome activity and NK cell function"

2. *Line 70 strain not strains.*

Since the second paragraph was too long, this sentence has been removed according to the reviewer's suggestion.

3. *Line 71 - "so called" is not necessary*

Since the second paragraph was too long, this sentence has been removed according to the reviewer's suggestion.

4. *Line 87 - delete the comma*

This sentence has been revised: "Alongside this canonical, caspase-1-dependent inflammasome pathway, the non-canonical caspase-11-dependent inflammasome (and its human orthologs, caspase-4 and -5) directly sense cytosolic lipopolysaccharide (LPS) and cleave GSDMD to trigger pyroptosis (Shi et al., 2015; Kayagaki et al., 2015; Yang et al., 2015; Vak et al., 2019)."

5. *Line 388 - Figure 1C is not appropriate for that sentence - probably for the previous sentence. However, the statement that IL-12 and IL-15 are involved in IFN γ production could do with a reference.*

The correction has been made and the reference has been added.

"However, a small decrease in IL-15 levels was observed in the IRF2^{KO} mice compared to WT mice (**Figure 1C**). Both IL-12 and IL-15 are key cytokines involved in regulating IFN- γ production (**Koka et al., 2004**)."

6. *Isn't figure S2D just the first two samples from Figure S2C?*

We acknowledge the need to clarify this point. Figure S2D (now Figure S2E) and Figure S2C (now Figure S2D) correspond to independent experiments performed at different MOIs. Specifically, Figure S2D (now Figure S2E) shows WT BMDMs infected with *F. novicida* at a MOI of 10, while Figure S2C (now Figure S2D) presents data from infections performed at a MOI of 1.

The text has been clarified: "As expected, no IFN- γ was produced either in the absence of BMDM infection or in the absence of SMCs (Figure **S2D-MOI1**; and Figure **S2E-MOI10**)."

7. *Line 461 - please insert infected "at MOI1 and 10" to indicate the difference between the difference between the supp data and the main figure. Took me a while to work out.*

The text has been clarified:

-“As expected, no IFN- γ was produced either in the absence of BMDM infection or in the absence of SMCs (Figure S2D-MOI1; and Figure S2E-MOI10). To assess the impact of IRF2 in macrophages on IFN- γ production, WT, IRF2^{KO} or GSDMD^{KO} BMDMs were infected and co-cultured with WT SMCs (Figure 3A-MOI1 and Figure S2F-MOI10)”.

-“We thus conducted additional experiments, co-culturing WT BMDMs with SMCs from WT, IRF2^{KO} or GSDMD^{KO} mice (Figure 3B-MOI1 and Figure S2G-MOI10)”.

8. *Line 510 Figure 4E-F The text says that other immune cell types were present in normal numbers. Just say ILC subsets rather than other immune cell types, because you haven't shown most cell types. You might need a split axis to actually see anything useful about the ILC numbers in this figure.*

The text has been clarified and the Figure has been updated according to the reviewer's recommendation.

“As shown Figure 4E-F, a similar reduction in NK cell numbers was observed in IRF2^{KO} mice prior to infection, whereas innate lymphoid cell (ILC) subsets were not statistically different compared to WT mice (Figure 4E-F)”.

Even though splitting the axis scale allowed a clearer visualization of ILC numbers, the differences between WT and IRF2^{KO} did not reach statistical significance.

9. *The writing is too small on some figures. Please check all figures for legibility.*

The text size has been increased to improve visibility in all the panels of Figure 4 and in panels E to F of Figure 6.

10. *Line 515-516 Perhaps it makes more sense to suggest that loss of IRF2 slows the release of NK cells from the bone marrow, leading to their increased number, rather than suggesting that it regulates NK cells at later stages, since in the next sentence you show profound differences in the bone marrow NK phenotype.*

The text has been revised in response to the reviewer's suggestion.

“Interestingly, the number of NK cells in the bone marrow of IRF2^{KO} mice was slightly increased compared to WT mice (Figure S3D), suggesting that loss of IRF2 may slow the exit of NK cells from the bone marrow”

11. *In general where there are trends that are not statistically significant, language needs to be modified to be less definite about the result.*

We agree with this point and have accordingly softened the language in the manuscript regarding Figures 1B, 1J, 4G, S3D, 4J, 6G, 6J and S1G.

Figure 1B: Upon systemic dissemination, higher colony-forming units (CFU) were observed in the lung, liver and spleen of IRF2^{KO} compared to WT mice. However, these differences were only statistically significant in the liver ($p=0.0042$), but not in the lung ($p=0.3180$) or spleen ($p=0.5515$) (Figure 1B).

Figure 1J: Our data indicate that IRF2 contributed to the regulation of GSDMD transcript and protein levels, both in naive (Figure 1H-I) and infected mice (Figure 1J-K) although the observed differences in GSDMD transcript levels did not reach statistical significance in the latter case.

Figure 4G: Furthermore, the maturation defect in peripheral IRF2^{KO} NK cells was also associated with reduced expression of GzmA and IL-18R, although not all differences were statistically significant (Figure 4G-H and Figure S3F-G) suggesting compromised cytotoxic activity and decreased responsiveness to IL-18, respectively.

Figure S3D: Interestingly, the number of NK cells in the bone marrow of IRF2^{KO} mice was slightly increased compared to WT mice (Figure S3D), suggesting that loss of IRF2 may slow the exit of NK cells from the bone marrow

Figure 4J: Additionally, both Eomes and Tbet expression levels were lower in NK cells in the spleen of IRF2^{KO} naive mice, compared to WT mice, with a statistically significant reduction observed only for Eomes (Figure 4J).

Figure 6G and 6J: Notably, granzyme A expression showed a trend toward upregulation (Figure 6G, blood and 6J, spleen), suggesting that, in the presence of IFN- γ , IRF2-deficient NK cells may retain some cytotoxic potential despite their immature status.

Figure S1G: Of note, GSDMD transcript levels were induced (3- to 4-fold increase depending on the organ) upon infection, with a statistically significant change observed only in the liver (Figure S1G).

12. Some references in your reference list are duplicated.

These errors have been corrected in the revised manuscript (Benaoudia et al 2019; Kayagaki et al 2019; Thygesen et al 2019).

13. Figure legends need some editing to ensure things are written conventionally.

Line 745-746; It is conventional to start with a description of the panels in alphabetic order.

The figure legends have all been verified, and those for Figures 1, 5 and 6 have been revised accordingly:

Figure 1: The following sentence has been corrected: "Mice of the indicated genotypes were subcutaneously (sc) injected with 7×10^2 (A) or $2,5 \times 10^3$ *F. novicida* (B-G, J-K) or with PBS (H-I)."

Figure 5: (A-G) Mice of the indicated genotypes were subcutaneously (s.c.) injected with 1,000 CFU (A) or 2,500 CFU (B-D, F-G), or intraperitoneally (i.p.) injected with 3 μ g of LPS per gram of body weight per mouse (E). (H-K) Mice of the indicated genotypes were s.c. injected with PBS.

Figure 6: (E-J) NK cell frequency (E-H), maturation (F-I), and GzmA expression (G-J) in the blood (E-G) and spleen (H-J) (results from 9 mice per group are shown).

14. Materials and Methods needs a bit of formatting eg some sections are written as multiple short paragraphs.

The formatting of the Materials and Methods section has been corrected. Notably, the subsections on Mice, animal infection and LPS injection were merged, as were those

on Immune cell phenotypic characterization and OMIQ analysis, Automated Western immunoblotting and conventional Western blotting, as well as ELISA and Luminex.

15. Line 634-635. Mariathasan should not be cited to say that GSDMD regulates IL-18 levels since the paper is from 2005 and they could not have tested the role of GSDMD.

That reference was removed

16. Line 379-378; "IFN- γ levels were decreased in the blood of IRF2 ko mice, pointing to a critical role of IRF2 in regulating IFN- γ " Since the role of IRF2 was later found to be quite indirect in regulating NK maturation, this might be better phrased.

The text has been revised:

IFN- γ levels were reduced in the blood of IRF2^{KO} mice, indicating that IRF2 plays a critical, albeit possibly indirect, role in regulating IFN- γ production

Referee #3:

General comment: *IRF2 is a transcription factor that has been previously shown to have a role in inflammasome activation through its regulation of caspase-4 and GSDMD levels. In this manuscript, the authors set out to understand the role of the transcription factor IRF2 in regulating in vivo responses to infection. The authors generated IRF2-deficient mice and find that they are as susceptible to Francisella novicida as GSDMD-deficient mice. They also find that IRF2-deficient mice also had a defect in IFN γ production, possibly due to a cell intrinsic role for IRF2 in regulating the number and maturation of NK cells. IRF2-deficient NK cells had a defect in expression of IL-18R and granzyme A. They were able to partially restore immune responses and control of infection by treatment of IRF2-deficient mice with recombinant IFN γ . Overall, the study is carefully designed, thorough, and well-written. These findings provide new insight into the role of IRF2 in regulating inflammasome activation as well as NK cell function. Furthermore, these findings provide a foundation for the potential use of IFN γ therapy in treating bacterial infections in patients with primary immunodeficiencies. This study will be of broad interest to basic researchers and clinicians in the fields of innate immunity, primary immunodeficiencies, and bacterial pathogenesis.*

We thank reviewer 3 for her/his careful evaluation of our work and for her/his constructive feedback.

Minor comments:

1. Fig. 1H: are there statistically significant differences in GSDMD transcript levels between WT and IRF2 ko mice? If not, the authors should reword the description of this data, as it currently says in the text that "We demonstrated that IRF2 positively regulated GSDMD transcript 398 and protein levels, both in naive (Figure 1F-G) and infected mice (Figure 1H-I, Figure 399 S1C)." In addition, the authors may want to

consider using ImageJ to quantify the protein levels of GSDMD in WT and IRF2 ko for figures 1G and I to determine whether there are indeed differences in GSDMD protein levels between WT and IRF2 ko mice.

We agree with this point and have accordingly softened the language in the manuscript. While GSDMD transcripts appear reduced *in vivo* following Francisella infection, the decrease is not statistically significant. The text has been revised to reflect this.

“Our data indicate that IRF2 contributed to the regulation of GSDMD transcript and protein levels, both in naive (Figure 1H-I) and infected mice (Figure 1J-K) although the observed differences in GSDMD transcript levels did not reach statistical significance in the latter case”.

To strengthen this observation, quantification of pro-GSDMD protein levels using ImageJ has been added (new Figures 1I and 1K), showing a significant decrease in IRF2^{KO} mice compared to WT following infection.

2. In some of the figures, such as Figure 5G-J, the authors show flow cytometry plots of histograms that they say is concatenate of 2 mice. Is it possible for the authors to do at least one more mouse for each of these analyses and then obtain statistical analysis?

We have included data from an additional mouse to have three mice per concatenates and perform statistical analyses. The results remain consistent with our initial findings. New Figures 5J–K present a representative experiment, which has been independently repeated three times with the corresponding statistical analyses on the right panels.

References:

Belhocine K & Monack DM (2012) Francisella infection triggers activation of the AIM2 inflammasome in murine dendritic cells. *Cell Microbiol* 14: 71–80

Fernandes-Alnemri T, Yu JW, Juliana C, Solorzano L, Kang S, Wu J, Datta P, McCormick M, Huang L, McDermott E, *et al* (2010) The AIM2 inflammasome is critical for innate immunity to Francisella tularensis. *Nat Immunol* 11: 385–93

Hagar JA, Powell DA, Aachoui Y, Ernst RK & Miao EA (2013) Cytoplasmic LPS activates caspase-11: implications in TLR4-independent endotoxic shock. *Science* 341: 1250–3

He W, Wan H, Hu L, Chen P, Wang X, Huang Z, Yang Z-H, Zhong C-Q & Han J (2015) Gasdermin D is an executor of pyroptosis and required for interleukin-1 β secretion. *Cell Res* 25: 1285–1298

Johansson A, Celli J, Conlan W, Elkins KL, Forsman M, Keim PS, Larsson P, Manoil C, Nano FE, Petersen JM, *et al* (2010) Objections to the transfer of Francisella novicida to the subspecies rank of Francisella tularensis. *Int J Syst Evol Microbiol* 60: 1717–8; author reply 1718-1720

Kayagaki N, Stowe IB, Lee BL, O'Rourke K, Anderson K, Warming S, Cuellar T, Haley B, Roose-Girma M, Phung QT, *et al* (2015) Caspase-11 cleaves gasdermin D for non-canonical inflammasome signalling. *Nature* 526: 666–671

Koka R, Burkett P, Chien M, Chai S, Boone DL & Ma A Cutting Edge: Murine Dendritic Cells Require IL-15R to Prime NK Cells. *The Journal of Immunology*

Lagrange B, Benaoudia S, Wallet P, Magnotti F, Provost A, Michal F, Martin A, Di Lorenzo F, Py BF, Molinaro A, *et al* (2018) Human caspase-4 detects tetra-acylated LPS and cytosolic Francisella and functions differently from murine caspase-11. *Nat Commun* 9: 242

Mariathasan S, Weiss DS, Dixit VM & Monack DM (2005) Innate immunity against Francisella tularensis is dependent on the ASC/caspase-1 axis. *J Exp Med* 202: 1043–9

Meunier E, Wallet P, Dreier RF, Costanzo S, Anton L, Ruhl S, Dussurgey S, Dick MS, Kistner A, Rigard M, *et al* (2015) Guanylate-binding proteins promote activation of the AIM2 inflammasome during infection with Francisella novicida. *Nat Immunol* 16: 476–484

Nielsen CM, Wolf A-S, Goodier MR & Riley EM (2016) Synergy between Common γ Chain Family Cytokines and IL-18 Potentiates Innate and Adaptive Pathways of NK Cell Activation. *Front Immunol* 7

Otani T, Nakamura S, Toki M, Motoda R, Kurimoto M & Orita K (1999) Identification of IFN- γ -Producing Cells in IL-12/IL-18-Treated Mice. *Cellular Immunology* 198: 111–119

Pierini R, Juruj C, Perret M, Jones CL, Mangeot P, Weiss DS & Henry T (2012) AIM2/ASC triggers caspase-8-dependent apoptosis in Francisella-infected caspase-1-deficient macrophages. *Cell Death Differ* 19: 1709–21

Pierini R, Perret M, Djebali S, Juruj C, Michallet MC, Forster I, Marvel J, Walzer T & Henry T (2013) ASC Controls IFN-gamma Levels in an IL-18-Dependent Manner in Caspase-1-Deficient Mice Infected with Francisella novicida. *J Immunol* 191: 3847–57

Shi J, Zhao Y, Wang K, Shi X, Wang Y, Huang H, Zhuang Y, Cai T, Wang F & Shao F (2015) Cleavage of GSDMD by inflammatory caspases determines pyroptotic cell death. *Nature* 526: 660–665

Teixeira M, Py BF, Bosc C, Laubretton D, Moutin M-J, Marvel J, Flamant F & Markossian S (2018) Electroporation of mice zygotes with dual guide RNA/Cas9 complexes for simple and efficient cloning-free genome editing. *Sci Rep* 8: 474

Vanaja SK, Russo AJ, Behl B, Banerjee I, Yankova M, Deshmukh SD & Rathinam VAK (2016) Bacterial Outer Membrane Vesicles Mediate Cytosolic Localization of LPS and Caspase-11 Activation. *Cell* 165: 1106–1119

Vak R, Y Z & F S (2019) Innate immunity to intracellular LPS. *Nature immunology* 20

Yang J, Zhao Y & Shao F (2015) Non-canonical activation of inflammatory caspases by cytosolic LPS in innate immunity. *Curr Opin Immunol* 32C: 78–83

Zhu Q, Zheng M, Balakrishnan A, Karki R & Kanneganti T-D (2018) Gasdermin D Promotes AIM2 Inflammasome Activation and Is Required for Host Protection against Francisella novicida. *J Immunol* 201: 3662–3668.

Dear Dr. Bourdonnay,

Thank you for the submission of your revised manuscript to our editorial offices. I have now received the reports from the two referees that I asked to re-evaluate the study, you will find below. As you will see, the referees now fully support publication of your study in EMBO reports. Both referees have remaining concerns or suggestions to improve the manuscript, I ask you to address in a final revised manuscript. Please also provide a final p-b-p-response regarding the remaining referee points and the editorial requests below.

Editorial requests:

- Please provide the abstract written in present tense throughout.
- Please order the manuscript sections like this, using only these names:
Title page - Abstract (max. 175 words) - Keywords (up to five) - Introduction - Results - Discussion - Methods - Data availability section - Acknowledgements (please include here all the funding information) - Disclosure and Competing Interests Statement - References - Figure legends - Expanded View Figure legends

Thus, please fuse the 'summary' with the abstract, or use this for the synopsis blurb (see below).

- Please move the declaration regarding AI tools to the Methods section.
- Please provide individual production quality figure files as .eps, .tif, .jpg (one file per figure), of main figures and EV figures. Please upload these as separate, individual files upon re-submission.

The Expanded View format, which will be displayed in the main HTML of the paper in a collapsible format, has replaced the Supplementary information. You can submit up to 6 images as Expanded View. Please follow the nomenclature Figure EV1, Figure EV2 etc. The figure legend for these should be included in the main manuscript document file in a section called Expanded View Figure Legends after the main Figure Legends section.

<https://link.springer.com/journal/44319/submission-guidelines#cms-Revised-submissions>

- Please check again that the number "n" for how many independent experiments were performed, their nature (biological versus technical replicates), the bars and error bars (e.g. SEM, SD) and the test used to calculate p-values is indicated in the respective figure legends (main, EV and Appendix figures). Please also check that all the p-values are explained in the legend, and that these fit to those shown in the figure. Please provide statistical testing where applicable. Please avoid the phrase 'independent experiment' but clearly state if these were biological or technical replicates. Please also indicate (e.g. with n.s.) if testing was performed, but the differences are not significant. In case n=2, please show the data as separate datapoints without error bars and statistics. See also:

<https://link.springer.com/journal/44319/submission-guidelines#cms-Figure-and-data-presentation>

If n<5, please show single datapoints for diagrams. It seems that presently some diagrams miss the statistics, show only partial statistics, or miss the 'n.s.'. Moreover:

- Please note that the legend for figure 1 is not provided in the sequential manner. This needs to be rectified.
- Please note that the exact p values are not provided in the legends of figures 1a, b, d-h, k; 2b-d; 3a, b; 4a, g, j; 5a-g, j, k; 6b-d, e, h, k, l; supplemental figures 1e, g; 2b, f, g; 3a, b, d, e
- Please note that the box plots need to be defined in terms of minima, maxima, centre, bounds of box and whiskers, and percentile in the legends of figures 4c
- Please note that information related to n is missing in the legends of figures 1k; supplemental figures 3e; 4b
- Please note that the error bars are not defined in the legends of figures 4k, l; supplemental figure 4b.
- Please provide the author checklist as excel file (as originally downloaded) and completed.

- Please make sure that each figure panel is called out separately and that the panels are called out sequentially. Presently, it seems there are no separate callouts for panels 6I and 6J. Please check.

- Please confirm that for all Western blot panels (main, EV, or Appendix figures) the loading control was run on the same gel as the other proteins detected. Please note that we discourage comparisons between samples on different gels/blots, even if the samples derive from one experiment, as confounding factors reduce comparability. If unavoidable, the figure legend must state that the samples derive from the same experiment and that gels/blots were processed in parallel. If a 'representative' loading control is shown for multiple gels/blots, the intra-gel controls should be shown in the source data files, and the figure legends

should describe the data displayed accurately. See our author guidelines:

<https://link.springer.com/journal/44319/submission-guidelines#cms-Figure-and-data-presentation> (section 'Electrophoretic gels and blots').

- All Materials and Methods need to be described in the main text using our 'Structured Methods' format, which is required for all research articles. According to this format, the Methods section should include a Reagents and Tools Table (listing key reagents, experimental models, software, and relevant equipment and including their sources and relevant identifiers), uploaded as separate file, and a Methods section in which we encourage the authors to describe their methods using a step-by-step protocol format with bullet points, to facilitate the adoption of the methodologies across labs. More information on how to adhere to this format as well as downloadable templates (.doc) for the Reagents and Tools Table can be found in our author guidelines (section 'Structured Methods'):

<https://link.springer.com/journal/44319/submission-guidelines#cms-Manuscript-organisation-and-formatting>

- Thanks for providing the source data. Please upload this as one folder per main figure, grouping together all the files for this figure (and ZIPed together). Moreover, please do not include there the source data checklist, but upload it separately.

In addition, I would need from you uploaded separately:

I look forward to seeing the further revised version of your manuscript when it is ready. Please let me know if you have questions regarding the revision.

Best,

Referee #1:

I have reviewed the revised manuscript "IRF2 deficiency disrupts pyroptosis, NK cell interferon- γ production and resistance to Francisella". The following minor comments should be addressed.

1. Figure S1C. The WB lanes should be labelled (e.g. which lane is WT and which lane is KO?).
2. At present, the TNF difference shown in Figure 2E is very pronounced and misleading if these data are not "reproducible" as stated by the authors, especially for the Δ FPI strain. Since the authors now indicate that TNF data contain "variability between WT and IRF2KO samples, but these differences were inconsistent and lacked reproducibility", the new pooled data should be added to Figure 2E to avoid publishing irreproducible difference.
3. The sample size appears inconsistent for Figure 1A:
The authors response in the point-by-point: "...infected WT and IRF2KO mice (Figure 1A) with a total of 20 WT and 17 IRF2KO mice".

Figure 1A figure legend: "The data were collected as follows: A shows results from 15 WT and 17 IRF2KO mice, with data pooled from 3 independent experiments".

Referee #2:

The authors have done a great job to further improve the clarity of the paper and should be commended. My comments are either minor or just for the consideration of the authors for whether they want to make any small changes.

1. Line 219-220. It is only true that IL-15 alone had no effect at the lower MOI - there was some effect in the supplementary

data.

2. Figure 6 Panels E and H should say frequency not frequency

3. Regarding the interpretation of Fig 2 and Fig S2A and timing of release of IL-1b and IL-18: Cells with blocked pyroptosis due to lack of GSDMD will undergo apoptosis and there is abundant cleaved and active caspase-8 and caspase-3 in the cells. With secondary necrosis, everything will eventually be released when the cell membrane breaks down. Rather than inferring there is some differential regulation of release of IL-1b and IL-18 from essentially dead cells (which to me seems unlikely), perhaps it is more likely that when GSDMD or IRF3 are absent and death becomes apoptotic, IL-18 is subject to cleavage that prevents its detection. IL-18 would be exposed to several hours of proteolytic action in the apoptotic cell if pyroptosis is blocked and it looks like there is literature on IL-18 cleavage by caspase-3.

4. My earlier comment on possible alternative promoter usage for GSDMD was really for the authors' information - there is no need to include this speculation if you think IRF-1 is a much more likely explanation.

5. Figure 1I right hand panel for quantification of the western blot was put in in response to the reviewer 3 I believe. I don't think you should bother with quantification of a single blot. I would recommend leaving this out, in consultation with the editor.

Dear Dr. Bourdonnay,

Thank you for the submission of your revised manuscript to our editorial offices. I have now received the reports from the two referees that I asked to re-evaluate the study, you will find below. As you will see, the referees now fully support publication of your study in EMBO reports. Both referees have remaining concerns or suggestions to improve the manuscript, I ask you to address in a final revised manuscript. Please also provide a final p-b-p-response regarding the remaining referee points and the editorial requests below.

Editorial requests:

- Please provide the abstract written in present tense throughout.

The present tense has been used throughout the abstract.

"IRF2 plays an indirect role in inflammasome activation by regulating Caspase-4 and Gasdermin D (GSDMD) levels. However, the in vivo relevance of this regulatory circuit is unknown. We generate IRF2^{KO} mice and demonstrate that they are equally susceptible to Francisella novicida infection as GSDMD^{KO} mice. Interestingly, the phenotypes of IRF2^{KO} and GSDMD^{KO} mice diverge with respect to IFN- γ . Specifically, IRF2^{KO} mice exhibit a profound defect in IFN- γ production, which we attribute to an intrinsic role of IRF2 in regulating both the number and maturation of NK cells. IRF2^{KO} NK cells fail to express the antibacterial effectors IL-18R and Granzyme A, thereby impairing bacterial clearance. IFN- γ therapy partially restores immune responses in IRF2^{KO} mice and resistance to infection. These findings confirm IRF2 as a dual regulator of inflammasome activity and NK cell function, highlighting its pivotal role in innate immunity. Moreover, they underscore the potential of IFN- γ therapy as a promising treatment for severe infections in patients with primary immunodeficiencies affecting multiple immune pathways".

- Please order the manuscript sections like this, using only these names:

Title page - Abstract (max. 175 words) - Keywords (up to five) - Introduction - Results - Discussion - Methods - Data availability section - Acknowledgements (please include here all the funding information) - Disclosure and Competing Interests Statement - References - Figure legends - Expanded View Figure legends

The manuscript sections have been ordered as indicated.

Thus, please fuse the 'summary' with the abstract, or use this for the synopsis blurb (see below).

The "summary" has been used for the synopsis blurb.

- Please move the declaration regarding AI tools to the Methods section.

The declaration regarding AI tools has been moved to the Methods section.

- Please provide individual production quality figure files as .eps, .tif, .jpg (one file per figure), of main figures and EV figures. Please upload these as separate, individual files upon re-submission. The Expanded View format, which will be displayed in the main HTML of the paper in a collapsible format, has replaced the Supplementary information. You can submit up to 6 images as Expanded View. Please follow the nomenclature Figure EV1, Figure EV2 etc. The figure legend for these should be included in the main manuscript document file in a

section called *Expanded View Figure Legends* after the main *Figure Legends* section.

<https://link.springer.com/journal/44319/submission-guidelines#cms-Revised-submissions>

The main figures and EV figures have been uploaded as .EPS files and provided as separate files. There are 5 EV Figures. The manuscript sections have been ordered as indicated.

- Please check again that the number "n" for how many independent experiments were performed, their nature (biological versus technical replicates), the bars and error bars (e.g. SEM, SD) and the test used to calculate p-values is indicated in the respective figure legends (main, EV and Appendix figures). Please also check that all the p-values are explained in the legend, and that these fit to those shown in the figure. Please provide statistical testing where applicable. Please avoid the phrase 'independent experiment' but clearly state if these were biological or technical replicates. Please also indicate (e.g. with n.s.) if testing was performed, but the differences are not significant. In case n=2, please show the data as separate datapoints without error bars and statistics. See also:

<https://link.springer.com/journal/44319/submission-guidelines#cms-Figure-and-data-presentation>

If n<5, please show single datapoints for diagrams. It seems that presently some diagrams miss the statistics, show only partial statistics, or miss the 'n.s.'.

This sentence was added at the end of the Statistical Analysis section of the Materials and Methods:

“Whenever a statistical difference or p value is not shown, the comparison was either not statistically significant or not tested.”

I also verified that all n values and indications are accurate and specified the exact p values, and I believe everything is now accurate.

Moreover:

- Please note that the legend for figure 1 is not provided in the sequential manner. This needs to be rectified.

The legend has been changed:

Mice of the indicated genotypes were subcutaneously (sc) injected with F. novicida or PBS. (A) Mice injected with 7×10^2 F. novicida were monitored for survival twice daily. (B) Mice injected with 2.5×10^3 F. novicida were assessed for bacterial burden in the lung, liver and spleen at day 2 post-inoculation. (C-D) Serum cytokine levels were measured at day 2 post-inoculation by Luminex assay. (E-F) Serum cytokine levels were measured at day 2 post-inoculation by ELISA. (G) Serum cytokine levels were measured at day 2 post-inoculation by Luminex assay. (H) Naive mice injected with PBS were assessed for GSDMD transcript levels. (I) GSDMD, pro-caspase-1, and β -actin protein levels were assessed in naive mice. In panel I, samples derive from the same experiment and gels/blots were processed in parallel. (J) GSDMD transcript levels were assessed in the spleen of mice injected with 2.5×10^3 F.

novicida at 48 hours post-inoculation. (K) Full-length and cleaved GSDMD (GSDMDc), ASC, and β -actin protein levels were assessed in the spleen at 48 hours post-inoculation by Western blot.

- Please note that the exact *p* values are not provided in the legends of figures 1a, b, d-h, k; 2b-d; 3a, b; 4a, g, j; 5a-g, j, k; 6b-d, e, h, k, l; supplemental figures 1e, g; 2b, f, g; 3a, b, d, e. We have now added the exact *p* values to the legends of all indicated figures. For clarity, *p* values are listed at the end of each figure legend.

- Please note that the box plots need to be defined in terms of minima, maxima, centre, bounds of box and whiskers, and percentile in the legends of figures 4c

We added this sentence in the legend:

The embedded box plots indicate the median (central line), the 25th and 75th percentiles (box), and the minimum and maximum values (whiskers). Each dot represents an individual mouse.

- Please note that information related to *n* is missing in the legends of figures 1k; supplemental figures 3e; 4b

Figure 1 legend: (K) Blot shows GSDMD expression from 3 WT and 3 IRF2^{KO} mice.

Supplemental Figure 3: (E): E shows results from 5 WT and 4 IRF2^{KO} mice

Supplemental Figure 4: (B): Gene expression values were obtained from two independent publicly available BioGPS datasets.

- Please note that the error bars are not defined in the legends of figures 4k, l; supplemental figure 4b.

The error bars in Figures 4K, 4L, and Supplemental Figure 4B represent the standard error of the mean (SEM). I have now added this information to the figure legends for clarity.

Figure 4 legend: (A, E-L) Bars indicate the mean with SEM.

Figure S4B legend: Bars indicate the mean with SEM

- Please provide the author checklist as excel file (as originally downloaded) and completed.

The author checklist has been provided.

- Please make sure that each figure panel is called out separately and that the panels are called out sequentially. Presently, it seems there are no separate callouts for panels 6I and 6J. Please check.

Here is the revised version with separate panel callouts for panels 6I and 6J:

“Interestingly, despite this expansion, the cells maintained an immature phenotype, as shown in Figure 6F (blood) and Figure 6I (spleen). Notably, granzyme A expression showed a trend toward upregulation, as shown in Figure 6G (blood) and Figure 6J (spleen), suggesting that, in the presence of IFN- γ IRF2-deficient NK cells may retain some cytotoxic potential despite their immature status.

- Please confirm that for all Western blot panels (main, EV, or Appendix figures) the loading control was run on the same gel as the other proteins detected. Please note that we discourage comparisons between samples on different gels/blots, even if the samples derive from one experiment, as confounding factors reduce comparability. If unavoidable, the figure legend must state that the samples derive from the same experiment and that gels/blots

were processed in parallel. If a 'representative' loading control is shown for multiple gels/blots, the intra-gel controls should be shown in the source data files, and the figure legends should describe the data displayed accurately. See our author guidelines:

<https://link.springer.com/journal/44319/submission-guidelines#cms-Figure-and-data-presentation> (section 'Electrophoretic gels and blots').

The legend of Figure 1, panel I, now states that the samples derive from the same experiment and that the gels/blots were processed in parallel.

- All Materials and Methods need to be described in the main text using our 'Structured Methods' format, which is required for all research articles. According to this format, the Methods section should include a Reagents and Tools Table (listing key reagents, experimental models, software, and relevant equipment and including their sources and relevant identifiers), uploaded as separate file, and a Methods section in which we encourage the authors to describe their methods using a step-by-step protocol format with bullet points, to facilitate the adoption of the methodologies across labs. More information on how to adhere to this format as well as downloadable templates (.doc) for the Reagents and Tools Table can be found in our author guidelines (section 'Structured Methods'):

<https://link.springer.com/journal/44319/submission-guidelines#cms-Manuscript-organisation-and-formatting>

The Reagents and Tools Table has been made, as well as a Methods section.

- Thanks for providing the source data. Please upload this as one folder per main figure, grouping together all the files for this figure (and ZIPed together). Moreover, please do not include there the source data checklist, but upload it separately.

The source data has been separated from the source data checklist.

In addition, I would need from you uploaded separately:

- a short, two-sentence summary of the manuscript (not more than 35 words).

The 35-word summary has been uploaded as well as the key findings below.

IRF2 regulates inflammasome activity and is essential for NK cell function. IRF2^{KO} mice show reduced IFN- γ and defective NK responses, impairing bacterial clearance. IFN- γ therapy partly restores immunity, suggesting a potential treatment for related immunodeficiencies.

- two to four short (!) bullet points highlighting the key findings of your study (two lines each).

- IRF2 regulates GSDMD in vivo in both naïve and infected mice, revealing previously unrecognized control of inflammasome-related pathways.
- IRF2 is essential for NK cell maturation and function, as shown within an integrated infection model linking IRF2 to antibacterial NK effectors such as GzmA and IL-18R.
- IRF2 deficiency disrupts immune defense by reducing NK cell numbers and impairing effector expression, weakening resistance to *F. novicida*.
- IFN- γ supplementation restores GSDMD and improves NK cell responses, enhancing immunity even with predominantly immature NK cells.

- a schematic summary figure as separate file that provides a sketch of the major findings (not a data image) in jpeg or tiff format (with the exact width of 550 pixels and a height of not more than 400 pixels) that can be used as a visual synopsis on our website.

A schematic summary figure with TIFF format has been uploaded.

I look forward to seeing the further revised version of your manuscript when it is ready. Please let me know if you have questions regarding the revision.

Please use this link to submit your revision: <https://embor.msubmit.net/cgi-bin/main.plex>

Best,

Referee #1:

I have reviewed the revised manuscript "IRF2 deficiency disrupts pyroptosis, NK cell interferon- γ production and resistance to Francisella". The following minor comments should be addressed.

1. *Figure S1C. The WB lanes should be labelled (e.g. which lane is WT and which lane is KO?).*
WB lanes are now labeled.

2. *At present, the TNF difference shown in Figure 2E is very pronounced and misleading if these data are not "reproducible" as stated by the authors, especially for the Δ FPI strain. Since the authors now indicate that TNF data contain "variability between WT and IRF2KO samples, but these differences were inconsistent and lacked reproducibility", the new pooled data should be added to Figure 2E to avoid publishing irreproducible difference.*
Figure 2E has been changed

3. *The sample size appears inconsistent for Figure 1A:
The authors response in the point-by-point: "...infected WT and IRF2KO mice (Figure 1A) with a total of 20 WT and 17 IRF2KO mice".*

Figure 1A figure legend: "The data were collected as follows: A shows results from 15 WT and 17 IRF2KO mice, with data pooled from 3 independent experiments".

The figure legend is accurate, sorry about the discrepancy. 15 WT mice have been used.

Referee #2:

The authors have done a great job to further improve the clarity of the paper and should be commended. My comments are either minor or just for the consideration of the authors for

whether they want to make any small changes.

1. Line 219-220. It is only true that IL-15 alone had no effect at the lower MOI - there was some effect in the supplementary data.

The sentence has been revised "In contrast, supplementation with IL-15 alone had no effect on IFN- γ levels in both IRF2^{KO} and GSDMD^{KO} BMDM/WT SMCs co-cultures at the lower MOI."

2. Figure 6 Panels E and H should say frequency not frequence

The word "frequence" has been changed to "frequency"

3. Regarding the interpretation of Fig 2 and Fig S2A and timing of release of IL-1b and IL-18: Cells with blocked pyroptosis due to lack of GSDMD will undergo apoptosis and there is abundant cleaved and active caspase-8 and caspase-3 in the cells. With secondary necrosis, everything will eventually be released when the cell membrane breaks down. Rather than inferring there is some differential regulation of release of IL-1b and IL-18 from essentially dead cells (which to me seems unlikely), perhaps it is more likely that when GSDMD or IRF3 are absent and death becomes apoptotic, IL-18 is subject to cleavage that prevents its detection. IL-18 would be exposed to several hours of proteolytic action in the apoptotic cell if pyroptosis is blocked and it looks like there is literature on IL-18 cleavage by caspase-3.

The text has been revised to incorporate the degradation hypothesis:

These findings suggest that although both IL-1 β and IL-18 are regulated by IRF2 and GSDMD, their detectable levels (secretion or degradation) follow distinct temporal patterns during *Francisella* infection. In particular, at late time points of infection, IL-1 β levels appear to be less dependent on the IRF2-GSDMD axis than IL-18 release.

4. My earlier comment on possible alternative promoter usage for GSDMD was really for the authors' information - there is no need to include this speculation if you think IRF-1 is a much more likely explanation.

The sentence has been removed.

5. Figure 11 right hand panel for quantification of the western blot was put in in response to the reviewer 3 I believe. I don't think you should bother with quantification of a single blot. I would recommend leaving this out, in consultation with the editor.

We chose to keep the quantification in order to address the concern raised by Reviewer 3.

Dr. Emilie Bourdonnay
CIRI, 21 avenue Tony Garnier, 69007 Lyon France.
69007
France

Dear Dr. Bourdonnay,

Thank you for the submission of your final revised manuscript to our editorial offices. It now went through this and your final p-b-p-response and consider the remaining points of the referees and the editorial requests as adequately addressed.

I am thus very pleased to accept your manuscript for publication in the next available issue of EMBO reports. Thank you for your contribution to our journal.

You may qualify for financial assistance for your publication charges - either via a Springer Nature fully open access agreement or an EMBO initiative. Check your eligibility: <https://link.springer.com/journal/44319/how-to-publish-with-us>

Yours sincerely,

>>> Please note that it is EMBO Reports policy for the transcript of the editorial process (containing referee reports and your response letter) to be published as an online supplement to each paper. If you do NOT want this, you will need to inform the Editorial Office via email immediately. More information is available here: <https://link.springer.com/partners/embo-press/editorial-policies#Peer%20review>